# Dynamic tuning of optical absorbers for accelerated solar-thermal energy storage

Zhongyong Wang[1], Zhen Tong[2], Qinxian Ye[1], Hang Hu[1], Xiao Nie[1], Chen Yan[2], Wen Shang [1], Chengyi Song[1], Jianbo Wu[1], Jun Wang[3], Hua Bao[2], Peng Tao [1] & Tao Deng [1]

Currently, solar-thermal energy storage within phase-change materials relies on adding high thermal-conductivity fillers to improve the thermal-diffusion-based charging rate, which often leads to limited enhancement of charging speed and sacrificed energy storage capacity. Here we report the exploration of a magnetically enhanced photon-transport-based charging approach, which enables the dynamic tuning of the distribution of optical absorbers dispersed within phase-change materials, to simultaneously achieve fast charging rates, large phase-change enthalpy, and high solar-thermal energy conversion efficiency. Compared with conventional thermal charging, the optical charging strategy improves the charging rate by more than 270% and triples the amount of overall stored thermal energy. This superior performance results from the distinct step-by-step photon-transport charging mechanism and the increased latent heat storage through magnetic manipulation of the dynamic distribution of optical absorbers.

[1] State Key Laboratory of Metal Matrix Composites, School of Materials Science and Engineering, Shanghai Jiao Tong University, Shanghai 200240, China. [2] University of Michigan–Shanghai Jiao Tong University Joint Institute, Shanghai Jiao Tong University, Shanghai 200240, China. [3] A123 Systems Research Center, A123 Systems, LLC, Waltham, MA 02451, USA. Zhongyong Wang, Zhen Tong and Qinxian Ye contributed equally to this work. Correspondence and requests for materials should be addressed to P.T. (email: taopeng@sjtu.edu.cn) or to H.B. (email: hua.bao@sjtu.edu.cn) or to T.D. (email: dengtao@sjtu.edu.cn)

Among various energy conversion processes[1,2], solar-thermal technology[3–8] has emerged as an attractive way to harness solar energy, particularly for heat-related applications, due to its extraordinary high energy conversion efficiency[2], low cost[6], environmental friendliness, and more importantly, the distinct storage capability of solar-thermal energy[7–11]. Solar-thermal energy storage has been developed as one of the key technologies to overcome the intermittency of solar radiation and to enable important solar-thermal applications ranging from managing building temperatures[12] to directly driving various industrial processes such as solar drying, steam generation, distillation, pasteurization, and electricity generation[13–17]. Similar to other energy storage technologies, the widespread utilization of solar-thermal energy storage lies in energy harvesting efficiency, storage capacity, and cost of the storage systems[5–8].

Compared to sensible thermal storage, latent heat storage with phase-change materials (PCMs) has much larger (5–14 times) storage capacity per unit volume, which significantly shrinks the size of storage materials and thus considerably reduces the cost of storage systems[18,19]. So far, most of the reported solar-thermal PCM systems were charged through thermal-diffusion-based thermal charging (TC) (Fig. 1a), in which a solar receiver is used to absorb and convert the solar energy into thermal energy, and the thermal energy is then transferred inside the PCMs through thermal diffusion. However, the low thermal conductivity ($k_{PCM}$, 0.1–1 W m$^{-1}$ K$^{-1}$)[19] of PCMs severely impedes the effective thermal energy transfer between the solar receiver and the PCMs, and in turn results in a slow movement of the charging interface and thus a slow charging rate. The slow charging process not only limits the real-time energy harvesting and the overall power stored during the available charging periods[20,21], but also causes serious safety concerns[22]. Past efforts addressing this slow charging problem have focused on improving $k_{PCM}$ through the addition of inorganic high-$k$ fillers[23], and more recently through the use of carbon nanotube (CNT) sponges[24], graphite and graphene foams[25–33] with high concentrations. Such high loadings often sacrifice the advantageous

features of organic PCMs, such as the large energy storage capacity, light weight, and good processibility. The obtained composites also only showed moderate improvement in both the $k_{PCM}$ and the charging rates[24–33]. Moreover, these composites often displayed unstable dispersion and abrupt drop of $k_{PCM}$ during the solid/liquid phase transition[34].

Besides using solar receivers, there has been exploration on solar-thermal energy harvesting through direct solar illumination on the surface of the PCM composites filled with light-absorbing carbon nanomaterials[24,25,27,28,33] or organic dyes[35,36]. However, these composites were not specifically designed to facilitate the light penetration into the storage media, and thus they still relied on thermal diffusion to accomplish the charging process. We recently demonstrated the rapid charging of transparent sensible heat storage materials (gel wax) via direct solar illumination of the composites doped with plasmonic gold nanoparticles (NPs)[37]. In such charging process, the energy transfer relies on fast transportation of photons within the transparent media to excite photothermal converters to optically charge the storage materials. This photon-transport-based optical charging (OC) approach is efficient for optically transparent storage media and application of such OC strategy to the PCM systems was considered to be challenging due to the opaqueness of the PCM[37].

In this work, we demonstrate that such photon-transport-based OC can still be applicable to rapid charging of the non-transparent PCMs through a step-by-step charging mechanism (Fig. 1b). More importantly, we show that the charging rate can be further speeded up by dynamic tuning the distribution of the Fe$_3$O$_4$@graphene optical absorbers with magnetic field (Fig. 1c), which can overcome the attenuated photon transport in normal OC (Fig. 1b) and lead to ~3 times improvement of the charging rate for the PCM composites. In addition to unparalleled fast charging rates, the extremely low loading requirement from this approach fully retains the large energy storage capacity of the PCM matrices, offers excellent stability of thermophysical properties and helps achieving high photothermal energy storage efficiency. In particular, the dynamic tuning strategy can be easily applied to solar-thermal

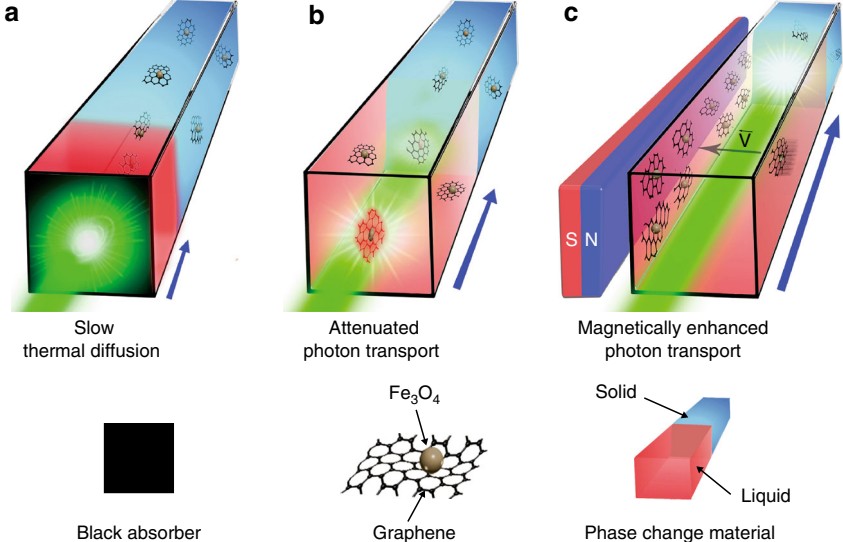

**Fig. 1** Solar-thermal energy storage within phase-change material composites. **a** Thermal-diffusion-based thermal charging. The incident light was converted into thermal energy by a black absorber film to charge thermal storage composites through thermal diffusion. Movement of charging interface was limited by slow thermal diffusion. **b** Photon-transport-based optical charging. The quick attenuation of incident light in the charged (liquid) state limits further light penetration. Movement of the charging interface was limited by attenuated photon transport. **c** Magnetically enhanced photon-transport-based optical charging. The dynamic removal of the optical absorbers from the optical charging path enables continuous photothermal conversion at the charging interface. Movement of charging interface was accelerated by the dynamic tuning of the distribution of the optical absorbers

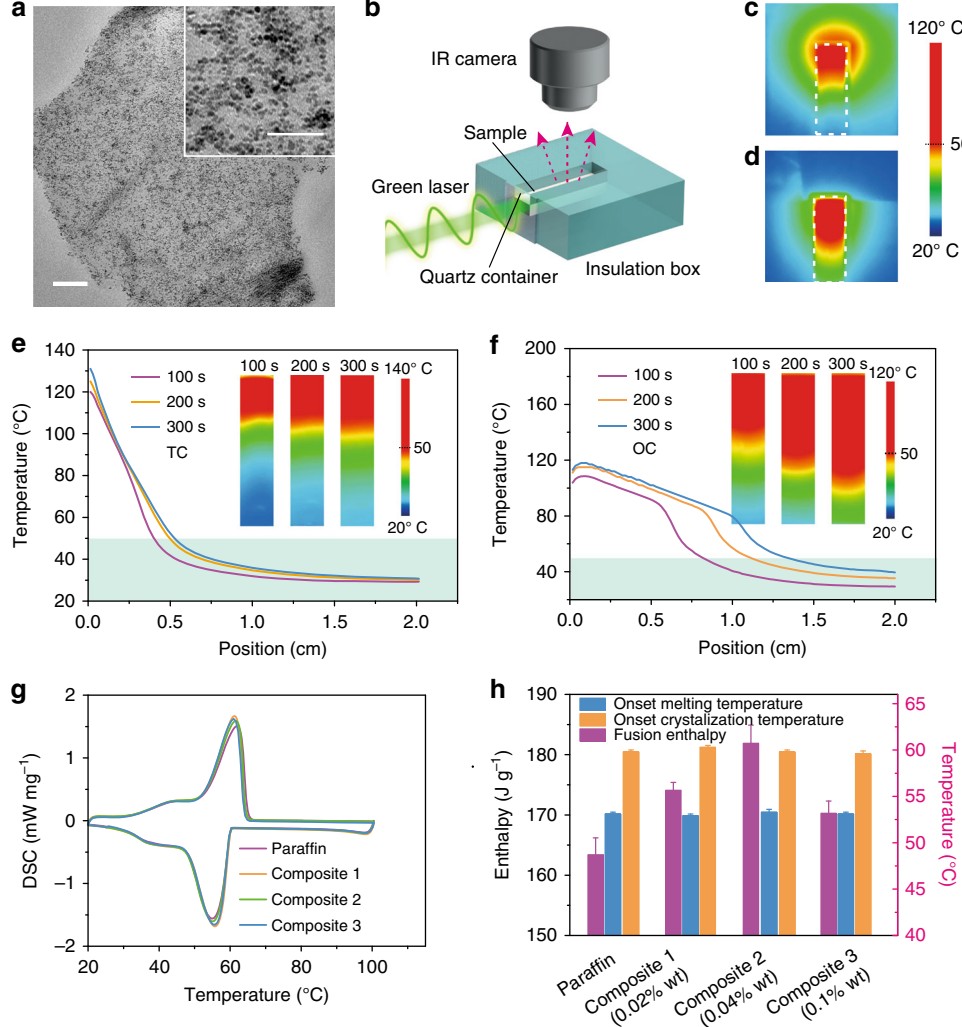

**Fig. 2** Charging of paraffin composites filled with $Fe_3O_4$@graphene nanoparticles. **a** Transmission electron microscopy (TEM) image of hybrid $Fe_3O_4$@graphene optical absorbers. The scale bar is 200 nm. The inset TEM image at a higher magnification shows the dense decoration of $Fe_3O_4$ nanoparticles on the surface of graphene sheets. The scale bar of the inset TEM image is 50 nm. **b** Schematic of experimental setup for charging of paraffin composites. A green laser (532 nm) with a power density of 4 W cm$^{-2}$ is used to illuminate a black aluminum foil in thermal charging (TC) mode or directly illuminate the surface of composites of paraffin-$Fe_3O_4$@graphene (0.02 wt%) in optical charging (OC) mode. **c, d** Infrared (IR) image of a thermally and optically charged sample after illumination for 100 s. The dashed line marks the boundary of the sample within the cuvette. **e, f** Time-sequential temperature distribution profiles of thermally charged sample and optically charged sample. The temperature profiles were extracted from the central line of the IR images. **g** Differential scanning calorimetry (DSC) curves of paraffin and paraffin composites with different loading of $Fe_3O_4$@graphene nanoparticles (composite 1: 0.02 wt%, composite 2: 0.04 wt%, composite 3: 0.1 wt%). **h** Comparison of fusion phase-change enthalpy, melting, and solidification temperatures of neat paraffin and paraffin-$Fe_3O_4$@graphene composites

harvesting with other different PCM systems or many other energy processes that involve optical absorption and conversion.

## Results

**Optical charging of PCM composites.** In this work we prepared the PCM composite model system by using the $Fe_3O_4$@graphene NPs as the photothermal converters, and paraffin wax, one of the common commercially available and widely used organic PCMs as the matrix. The $Fe_3O_4$@graphene hybrid NPs were used due to the good optical absorption of graphene and also the good magnetic response of $Fe_3O_4$ NPs. They were synthesized through direct nucleation and controlled growth of $Fe_3O_4$ NPs on the surfaces of graphene oxide (GO) sheets, and followed by in situ reduction with oleylamine (OLA)[38] (Supplementary Note 1). Transmission electron microscopy (TEM) observation shows

dense decoration of $Fe_3O_4$ NPs ($7 \pm 2$ nm) on the surface of the graphene sheets (Fig. 2a). X-ray diffraction (XRD) patterns show the diffraction peaks of $Fe_3O_4$ and the disappearance of GO characteristic (001) peak in the hybrid NPs (Supplementary Fig. 1), indicating the successful reduction of GO into graphene. This transformation is further confirmed by the increased Raman spectral intensity ratio of the D band and G band ($I_D/I_G$) from the GO to the hybrid NPs[39] (Supplementary Fig. 1). The X-ray photoelectron spectroscopy (XPS) spectrum of Fe 2p level shows the existence of both $Fe^{2+}$ and $Fe^{3+}$ binding states in the hybrid NPs indicating the formation of $Fe_3O_4$ NPs[40] (Supplementary Fig. 2). The XPS spectrum of C 1s level also shows the decrease of C–O and C=O binding peak intensity from GO to $Fe_3O_4$@-graphene NPs (Supplementary Fig. 2). After the reduction process, the black-colored $Fe_3O_4$@graphene NPs thus can efficiently absorb light and convert it into heat. Meanwhile, owing

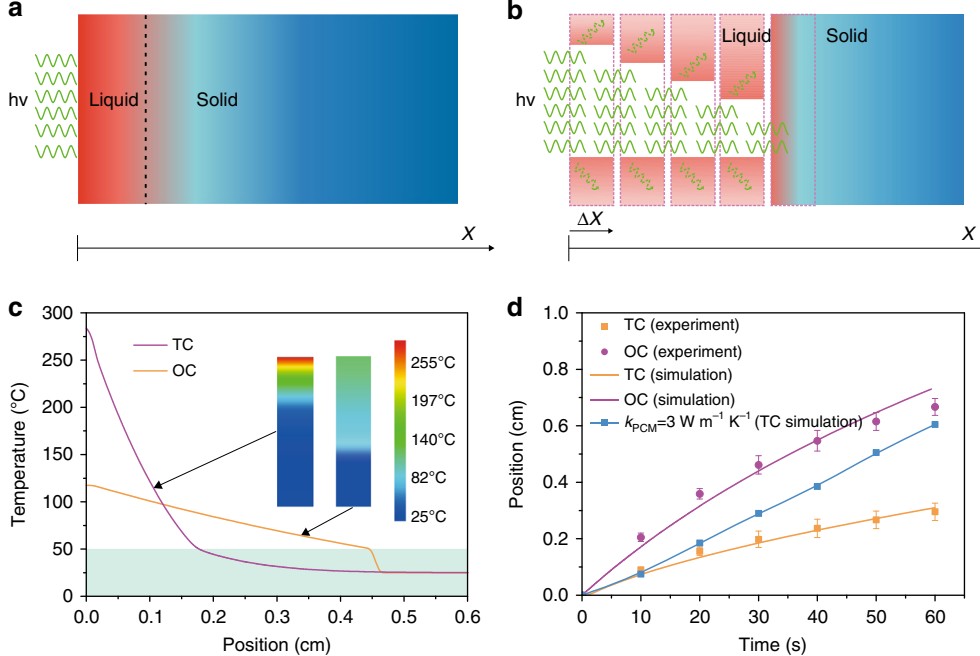

**Fig. 3** Theoretical modeling of the charging of paraffin-Fe$_3$O$_4$@graphene composites. **a** Simulation schematic of thermal charging (TC) process. A constant heat flux converted from laser illumination is applied as the fixed heat source to charge PCM via thermal diffusion. **b** Simulation schematic of optical charging (OC) process for the non-transparent solid paraffin composites. The whole sample is divided into numerous domains and is melted step by step. **c** Simulated temperature distribution profiles after charging for 30 s. **d** Simulation of solid/liquid interface propagation and comparison to experiments. The solid/liquid interface movement of thermally charged paraffin composites with a hypothetical thermal conductivity of 3 W m$^{-1}$ K$^{-1}$ (blue line) was simulated for comparison

to surface passivation with OLA ligands the synthesized Fe$_3$O$_4$@graphene NPs can be uniformly dispersed within paraffin to prepare PCM composites.

To explore the application of OC to this PCM system, light transmission of the PCM composites under both solid state and liquid state was measured. With 0.02 wt% of Fe$_3$O$_4$@graphene NPs, the paraffin composite (with a thickness of 3 mm) has a transmittance of about 80% and about 30% for the melted liquid state and the solid state, respectively (Supplementary Fig. 1). The increased transmission of light in the liquid state of the composite material provides the possibility for the penetration of the charging light through the melted PCM and thus enables the OC process. We first compared the charging behavior of these PCM composites under both the TC mode and the OC mode. The charging process of the paraffin composites was investigated by monitoring the temperature evolution of the sample placed within a quartz container that is surrounded by the thermal-insulating polystyrene foam with an infrared (IR) camera (Fig. 2b). Under the TC mode, a black aluminum absorber in direct contact with the sample was used as the light receiver to charge paraffin. Limited by the low thermal conductivity of the paraffin composite, the converted thermal energy is accumulated at the conversion front and tends to be easily lost to the surrounding. The IR image in Fig. 2c shows a large heat-affected zone surrounding the conversion front. Under the OC mode, the thermal energy converted by the hybrid Fe$_3$O$_4$@graphene NPs can be timely stored within the paraffin matrix, thus the associated heat loss from the optically charged PCMs is reduced (Fig. 2d).

Figure 2e presents that the thermally charged sample has a large temperature gradient near the absorber and only about 1/4 of the solid paraffin transits into liquid state. By prolonging the charging time from 100 to 300 s, the temperature profile has relatively small change, which shows that the converted thermal

energy cannot be effectively stored into the PCM. By comparison, in the optically charged sample the temperature is more evenly distributed and much larger portion of the paraffin composite was melted into liquid to store the solar-thermal energy (Fig. 2f). As delineated by the green boundaries in Fig. 2e, f, the position of the charging interface was tracked by intersecting the temperature distribution profile with the onset melting temperature of the PCM (50 °C). With 0.02 wt% loading of Fe$_3$O$_4$@graphene NPs, the advancement of the charging interface within the same 300 s charging duration is increased by 148%. The thermal conductivity of paraffin composites with 0.02 wt% of hybrid NPs was measured by a transient hot bridge analyzer to be nearly the same as the neat paraffin (0.23 W m$^{-1}$ K$^{-1}$). Thermal-conductivity enhancement thus has negligible contribution to the observed higher charging rates in the optically charged samples.

Differential scanning calorimetry (DSC) curves in Fig. 2g reveal that the composite samples have almost the same melting temperature and solidification temperature as the neat paraffin wax owing to the low loading of the hybrid NPs and also the surface capping of NPs with OLA that improves the compatibility between the hybrid NPs and the matrix, and minimizes potential interruption of the movement of alkane chains in paraffin wax. Unlike thermal-conductivity enhancement approach, the loading requirement in OC process is at least one order of magnitude lower. Such extremely low loading is beneficial for achieving high energy storage capacity, as the phase-change enthalpy of the composite is strongly dependent on the volume fraction of the organic PCM[24,26,30,32]. In contrast to the widely reported serious reduction of thermal storage capacity after addition of fillers, it was found that the solid-liquid fusion enthalpy of the neat paraffin (163 J g$^{-1}$) was even increased by more than 10% in the composite sample with 0.04 wt% of hybrid NPs (182 J g$^{-1}$) (Fig. 2h). When the particle loading was higher than 0.1 wt%,

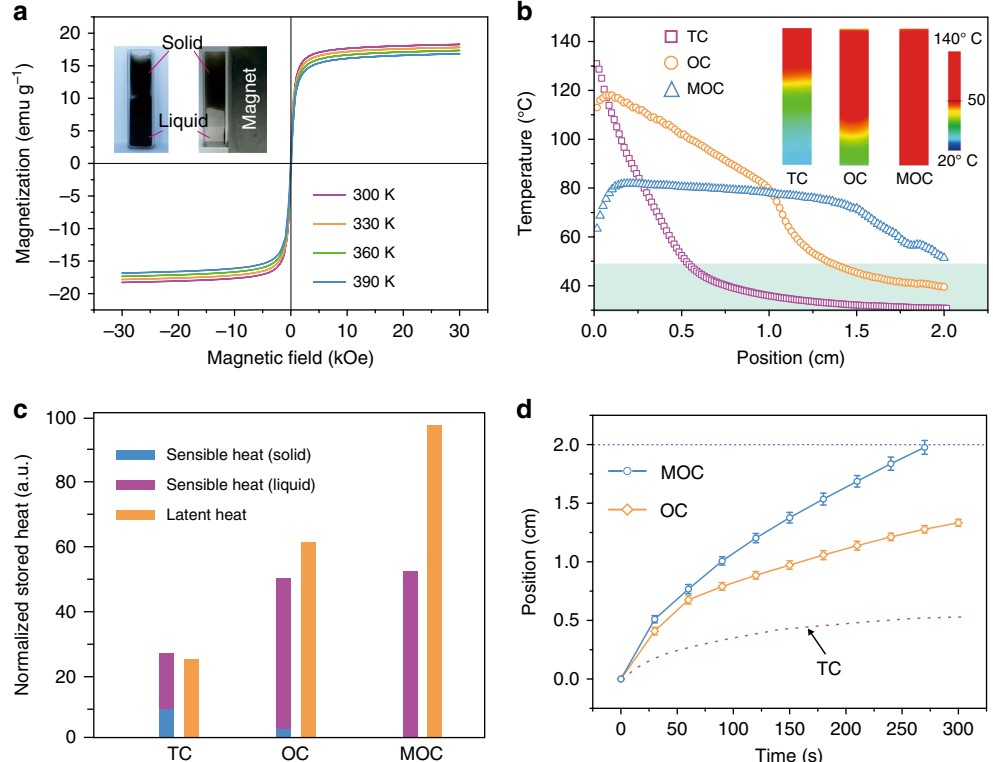

**Fig. 4** Magnetically accelerated charging of paraffin composites. **a** Magnetization curves of $Fe_3O_4$@graphene nanoparticles (NPs) tested within the temperature range of 300–390 K. The inset images show the effective removal of $Fe_3O_4$@graphene NPs from the homogenous dispersion in the melted composite to the side wall under the attraction from the magnet. **b** Temperature distribution profiles of paraffin composites (0.02 wt%) under thermal charging (TC), optical charging (OC), and magnetically enhanced optical charging (MOC) mode after charging for 270 s. The inset infrared (IR) images show faster charging rates and more uniform temperature distribution under the MOC mode. **c** Comparison of the stored sensible heat and latent heat under TC, OC, and MOC modes. **d** Movement of the solid/liquid charging interface as a function of charging time. The top dashed line marks the sample boundary

we observed the decrease of the latent heat with the increase of the particle loading, which has been more commonly reported[26,32]. When the particle loading was < 0.1 wt%, we observed the increase of the latent heat as the particle loading increased. Recently there have been several reports of the increase of the latent heat as the loading increases, especially with carbon-based nanofillers[24,41–47]. The favorable intermolecular interaction between paraffin and carbon nanofillers was proposed as the possible reason to account for the increase of the latent heat, but the exact mechanism is still under debate[24,43,44].

To gain theoretical insights into the observed different charging behaviors, we used a classical analytic method that does not consider heat losses[30,48] to model the TC and OC process (Supplementary Note 2). Under the TC mode, a constant heat flux is fixed in the front portion to induce temperature rise and phase transition in PCM (Fig. 3a). In the OC model, we proposed a step-by-step charging mechanism, in which the phase-change domain is divided into many small slices with a step size of $\Delta x$ (Fig. 3b). The thin solid PCM slice is melted by the photothermal conversion of the $Fe_3O_4$@graphene NPs within the PCM. As soon as the solid PCM slice is melted to liquid, the incident laser beam can pass through the relatively transparent liquid PCM composite slice and move forward to melt the next solid PCM slice. In this case, a moving heat source is applied on the solid PCM during the simulation.

The simulated temperature distribution profile in Fig. 3c reveals a large temperature gradient in the melted liquid region under the TC mode because most of the energy is absorbed by the PCM as sensible heat in the front liquid region rather than latent

heat. Due to the low $k_{PCM}$, the converted heat cannot be timely transferred to the solid paraffin therefore resulting in a continuous temperature rise of the melted paraffin liquid. In contrast, for the step-by-step OC process, most of the energy has been used to induce the desired phase change of solid PCMs, and only a small amount of optical energy input is absorbed by the liquid region and stored as sensible heat. This step-by-step charging mechanism leads to significantly increased latent heat storage of the PCM and much more uniform temperature distribution within the PCM. To have a clear comparison of the physics of the TC mode and OC mode, we did not include the heat loss in Fig. 3. With the consideration of the heat loss, the calculated temperature distribution is closer to the measured temperature distribution (Supplementary Note 3).

Based on energy conservation principle, i.e., the solar-thermal energy input equals the sensible and latent heat stored within the PCM, we also simulated the propagation of solid/liquid interface as a function of charging time. To simulate OC process, appropriate selection of step size is critical. By trying with different step sizes, it was found that when $\Delta x$ approaches 0.01 cm the simulation results converge and further decreasing $\Delta x$ does not change much the simulated interface movement (Supplementary Fig. 3a). As shown by Fig. 3d, the simulation results confirm the much faster charging rates in the OC mode and match well with experimental data within the initial 60 s. With prolonged charging time, the theoretical model tends to overestimate the movement of charging interface and tempera-ture due to the increase of heat loss as the temperature of the composites increases[30,47,48] (Supplementary Fig. 3b). With a

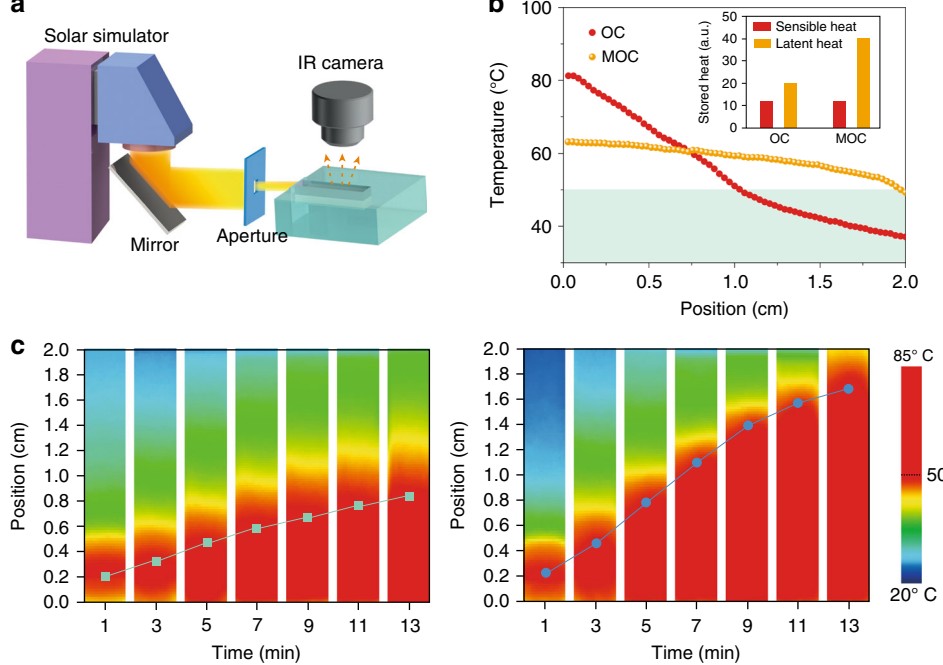

**Fig. 5** Solar-thermal energy harvesting via dynamic tuning-based optical charging of paraffin composites. **a** Schematic illustration of direct solar-thermal energy harvest. **b** Temperature profiles of charged composites (with 0.1 wt% of the hybrid $Fe_3O_4$@graphene nanoparticles) under optical charging (OC) and magnetically enhanced optical charging (MOC) mode after charging for 15 min. The inset figure compares the corresponding portion of sensible and latent heat storage. **c** Time-dependent solar charging interface movement under OC (left) and MOC (right) mode

hypothetical high $k_{PCM}$ of $3\,W\,m^{-1}\cdot K^{-1}$, which is close to the highest $k_{PCM}$ ($3.5\,W\,m^{-1}\cdot K^{-1}$) reported so far for organic PCM composites without severely decreasing the storage capacity[30,32], the calculated charging rate in TC mode is still lower than that in the OC mode. Whereas, both the measured and simulated interface movement in Fig. 3d shows that the charging rate in OC mode gradually slows down as the charging process further proceeds. As schemed by Fig. 3b, the reduction in charging rate can be attributed to the fact that as more PCM composites are melted, the light absorption in liquid PCM region increases, the amount of optical energy accessible for photothermal conversion in solid PCM decreases. This attenuation effect becomes more pronounced with increasing loading of NPs, which explains the slowed charging observed in the 0.4 wt% sample (Supplementary Fig. 4). It can be expected that further increasing the loading of optical absorbers in the PCM matrix, such as the composites reported previously[24–28,33], would finally result in slow charging similar to the TC mode since all the light is absorbed by the optical absorbers at the front portion of the samples and further penetration of light is impeded for the OC mode.

**Optical charging with dynamic tuning of the distribution of optical absorbers**. Theoretical analysis of the OC mechanism reveals that by minimizing the light absorption in the melted liquid-phase PCMs more light can reach the liquid–solid charging interface after its penetration through the liquid phase, and the charging rates should be further improved. One way to minimize the light absorption in the liquid-phase PCMs is to remove the optical absorbers, in our system the hybrid $Fe_3O_4$@graphene NPs, from the liquid-phase PCMs during the charging process. In this work, we introduced external magnetic field to manipulate the dynamic distribution of the optical absorbers during the OC process by taking advantage of the quick magnetic response from the $Fe_3O_4$ NPs on the graphene surfaces. Figure 4a presents that the hybrid $Fe_3O_4$@graphene NPs have exhibited stable

magnetization behavior within the tested temperature range of 300–390 K. The zero-field cooling (ZFC) and field-cooling (FC) magnetization curves indicate that the $Fe_3O_4$@graphene NPs have a superparamagnetic blocking temperature of 110 K that is associated with the superparamagnetic $Fe_3O_4$ NPs[40,49] (Supplementary Fig. 5). Once the PCM composites are melted, within 6 s the hybrid NPs are fully attracted to the side wall by a permanent magnet with a magnetic field strength of about 0.3 Tesla. The timely removal of $Fe_3O_4$@graphene optical absorbers in the melted paraffin from the light path would help further advance the solid/liquid charging interface of the composite.

Figure 4b compares the charging process of the PCM composites with 0.02 wt% loading of $Fe_3O_4$@graphene NPs under three different modes: TC, OC, and magnetically enhanced OC (MOC) with the magnet placed at the side wall. The IR images in Fig. 4b present that attracting the hybrid NPs in the melted paraffin to the side wall significantly accelerates the propagation of the charging interface. Simultaneously, the overall temperature distribution within the charged PCMs becomes more uniform, thus effectively suppressing the potential overheating in the TC mode. The difference in the charging mechanisms led to the difference in the temperature profiles, which also resulted in the differences in the temperature-dependent heat losses that include the radiation and convection heat losses at the front and top surfaces, and the conduction heat losses to the side walls of the container (Supplementary Note 3 and Supplementary Fig. 6). Based on the temperature distribution curves, we can also estimate the portion of sensible and latent heat storage within the charged samples (Supplementary Note 4 and Supplementary Fig. 7). Figure 4c shows that because of the strong heat localization in the TC mode, the amount of stored sensible heat is even larger than the latent heat and about 1/3 of sensible heat is stored within the solid PCM. By comparison, OC and MOC modes have shown significantly larger melted zone and thus larger latent heat storage than the TC mode, and the overall stored thermal energy was increased by 118% and 195% for OC

and MOC processes, respectively. For the desired latent heat storage, the MOC mode has led to ~300% increase over the TC mode and about 60% increase over the OC mode. Figure 4d indicates that the MOC mode has a higher charging rate from the beginning and the enhancement effect becomes more pronounced with longer charging duration. After charging for 270 s, the MOC, OC, and TC modes lead to an advancement of solid/liquid interface by 2.0, 1.24, and 0.53 cm, respectively (Fig. 4c), meaning more than 60% and 270% increase of the time-average charging rate by the MOC mode over the OC and TC mode. The heat loss analysis (Supplementary Figs. 6 and 8) shows that the heat loss in MOC mode is much smaller than that in TC and OC mode during the charging process. With the consideration of the heat losses, the simulated temperature distribution profiles are closer to the measured temperature distributions in Fig. 4b (Supplementary Fig. 9).

**Solar-thermal energy harvest**. The hybrid $Fe_3O_4$@graphene NPs have exhibited broad-band absorption of solar light ranging from 300 to 2000 nm (Supplementary Fig. 10). Such effective absorption of solar radiation enables the paraffin-$Fe_3O_4$@graphene composites to be directly employed to harvest and store solar-thermal energy as well. The concentrated solar light (1 W cm$^{-2}$) from a solar simulator was directly shed on the composites to harvest solar-thermal energy (Fig. 5a). The temperature distribution profiles in Fig. 5b show that under the same solar illumination the average charging rate of MOC mode is twice of that of OC mode for the PCM composite with 0.1 wt% of hybrid NPs. The overall stored energy or the instant energy harvesting efficiency was increased by 64.3% in the MOC mode. The time-sequential IR images in Fig. 5c show the gradual movement of the charging interface under the OC mode and faster advancement under the MOC mode. Furthermore, upon completing the charging process under the MOC mode, the dispersion state of hybrid NPs within PCMs can be easily recovered to the initial homogeneous mixing state by simply applying the alternating magnetic field (Supplementary Fig. 11).

To compare the charging performance of the PCM composite systems developed in this work to other PCM composite systems, we measured the temperature rising curves of thin samples (with a thickness ~3 mm, Supplementary Note 5). Under MOC charging mode, the composite sample with 0.1 wt% NPs has shown a high energy storage efficiency of 92.4% (Supplementary Fig. 12), which is much higher than the reported efficiency of 40–60% in paraffin CNT sponge systems[24] and is comparable to the highest value reported so far but with much lower loading concentrations[25,27,33,35,37,40]. The achieved high storage efficiency, for one thing, should be related to the excellent solar absorption capability and high photothermal conversion efficiency of graphene[50]. More importantly, our approach intrinsically offers higher charging rates, much more uniform temperature distribution, and less heat loss from the charged composites, thus benefiting for achieving higher storage efficiency.

Additionally, the amount of fillers has been minimized in our system and their surfaces have been modified with matrix compatible ligands. The OLA-capped $Fe_3O_4$@graphene hybrid NPs have shown excellent dispersion stability against agglomeration and precipitation. The good dispersion enables the effective solar absorption with the extremely low loading concentration of the absorbers. The PCM composites demonstrated stable photothermal energy storage performance in terms of phase transition enthalpy, photothermal storage efficiency, transition temperature, thermal cycling stability as evidenced by the overlapped DSC curves after repeated heating/cooling tests and

repeated MOC process for 100 cycles (Supplementary Fig. 13). Other advantages associated with the low filler-loading requirement for the MOC process include good processability, low cost, and the reduced weight of overall solar-thermal storage system.

## Discussion

Effective tuning of the distribution of photothermal converters within melted PCM is the key to realizing accelerated charging process. During solar charging experiment, it was found that the $Fe_3O_4$@graphene NP loading up to 0.1 wt% could be effectively attracted to the side wall of a Petri dish with a diameter of ~4 cm. Increasing loading higher than 0.1 wt% leads to lowering the heat of fusion after the initial enhancement of heat fusion with loading lower than 0.1 wt%. Further increasing NP loading would also increase the viscosity of the melted PCM composites[43,51], which would limit the movement of the hybrid NPs. Developing hybrid photothermal converters with stronger magnetic response and applying stronger magnets such as electromagnets would enable effective manipulation of the absorbers within large size charging systems. In addition to the low-temperature applications by using paraffin as the PCM, the MOC approach could also be applicable for rapid high-temperature solar-thermal energy charging with the development of high-temperature dual-functional solar-thermal converters such as low-cost carbon-based absorber and using molten salt as the PCM matrix. The demonstrated advantageous features including extremely low loading requirement, effective tuning distribution with external magnets, and facile preparation process hold the promise of scaling up the charging system for practical solar-thermal applications.

In this work, we have shown that dynamic tuning the distribution of the optical absorbers realizes the simultaneous achievement of fast charging rates, large phase-change enthalpy and high solar-thermal energy conversion efficiency for the thermal storage PCMs. The validated step-by-step theoretical model enables the photon-transport-based charging in the non-transparent storage media and the application of magnetic field to manipulate the dynamic distribution of optical absorbers during the phase-change process to further improve the charging rate. Tunable distribution of the optical absorbers offers another route to improve solar-thermal harvesting performance and this strategy also holds the great potential for many other applications that involve the absorption and conversion of photonic energy.

## Methods

**Preparation and charging of PCM composites**. Oleylamine-capped $Fe_3O_4$@-graphene hybrid NPs were synthesized as optical absorbers and mixed with melted paraffin wax followed by removal of the chloroform dispersing solvent to prepare PCM composites with stable dispersion during photothermal conversion. The obtained solid PCM composites were then cut into a cuboid (0.4 cm × 0.4 cm × 2 cm) and loaded into a quartz container. The whole sample was thermally insulated by polymer foams except the front light incident side and the top temperature monitoring window. The sample was thermally charged by using a black aluminum foil (0.4 cm × 0.4 cm × 50 μm, Thorlabs) as the solar absorber or optically charged by direct illumination of a green laser (532 nm) with a beam diameter of 5 mm or simulated solar light onto the sample. To realize magnetically enhanced optical charging, a magnet was placed at the sidewall of the container before illumination. The temperature evolution of the whole samples was recorded by a thermal infrared camera (T620, FLIR Systems Inc.) operated under a video mode from the top surface.

**Characterization and property measurement**. The morphology and size of as-synthesized $Fe_3O_4$@graphene NPs were observed with a TEM (Tecnai G2 Spirit Biotwin, FEI) operating at 120 KV. XRD patterns were collected by an X-ray polycrystalline diffractometer (D8 Advance, Bruker). Raman spectra were collected by a dispersive Raman microscope (Senterra R200-L, Bruker). XPS spectra were measured by an X-ray photoelectron spectrometer (AXIS, UltraDLD). Magnetization curves of the $Fe_3O_4$@graphene NPs were measured by a physical property measurement system (PPMS-9T, Quantum Design). Thermal conductivity of paraffin and paraffin-$Fe_3O_4$@graphene composite samples was measured by a transient hot bridge analyzer (Linseis THB-1). Optical spectra were measured by a

UV-Vis spectrometer (Lambda 950, PerkinElmer) equipped with an integrating sphere. TGA and DSC analysis was conducted in a differential scanning calorimeter (Netzsch 204 F1) and a thermo gravimetric analyzer (Pyris 1) under nitrogen atmosphere with a heating and cooling rate of 5 °C min$^{-1}$, respectively.

**Theoretical simulations**. Thermal charging process was modeled as one-dimensional heat diffusion problem. The dynamic temperature distribution of paraffin composites $T(x, t)$ in liquid phase and solid phase was obtained by analytically solving the standard heat diffusion equations in liquid region and solid region. To simulate optical charging process, we developed a step-by-step model. As shown in Fig. 3b, the simulation domain was divided into numerous slices with a step size of $\Delta x$. During optical charging process, the whole PCM composites were melted slice by slice. After illumination for certain duration, only single slice was melt. After deducting the liquid absorption loss, the remained optical energy further induced uniform heating and melting of the next slice. The temperature in liquid region can be obtained by solving the energy conservation equation between energy input from the absorbed optical energy in the control volume and the energy needed to melt the solid PCM in the control volume, including both sensible heat and latent heat. Both the theoretical simulation without considering the heat losses and the theoretical simulation considering the heat losses are included in the Supplementary Information. All the experimental parameters and the physical properties of the materials used in the simulation are listed in Supplementary Table 1.

**Data availability**. The data that support the findings of this study are available within the paper and its Supplementary Information File, or from the corresponding authors on request.

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

## Acknowledgements

The work was supported by National Natural Science Foundation of China (51403127, 51521004, 51420105009, 21401129, and 51676121), National Key R&D Program of China (No. 2017YFB0406100), "Chen Guang" project from Shanghai Municipal Education Commission and Shanghai Education Development Foundation (Grant No. 15CG06), the Fundamental Research Funds for the Central Universities and the Zhi-Yuan Endowed fund from Shanghai Jiao Tong University. We also thank all the reviewers for their helpful comments in improving the manuscript.

## Author contributions

T.D., P.T., and H.B. conceived, planned, and supervised this study. Z.W., Q.Y., H.H., and X.N. carried out experimental work. Z.T. and C.Y. performed numerical simulation analysis. Z.W., Z.T., and Q.Y. contributed equally to this work. All authors discussed the results and contributed to the writing of the manuscript.

## Additional information

**Competing interests:** The authors declare no competing financial interests.

