## [Peer Review File · Nature Communications]

Reviewers' Comments:

Reviewer #1 (Remarks to the Author)

The manuscript described dynamic tuning the distribution of Fe₃O₄@graphene in paraffin composites to achieve fast charging rates, large phase change enthalpy and high solar-thermal energy conversion efficiency. This research work is highly innovative, and will be of interest to others in the community. The manuscript can be published after the following questions are solved.

1. The characterization of Fe₃O₄@graphene needs to be supplemented and completed.
(1) The authors claim that the fabricated nanoparticles are magnetite but there is no clear evidence. XRD data themselves do not allow to make this conclusion because maghemite and magnetite are identical for XRD. The XPS spectrum of Fe₃O₄ needs to be supplemented.
(2) The authors should provide more evidence for the reduction of GO into graphene. For example, the XPS spectrum of C 1s for GO and Fe₃O₄@graphene.
(3) M-H curve measured from -30 kOe to 30 kOe cannot determine the superparamagnetic behavior. The magnified hysteresis curve or ZFC/FC curve may be adopted to characterize the magnetic behavior.

The authors can refer to the following literatures: Journal of Materials Chemistry A 2017, 5, 958-968

2. Since the saturation magnetization of Fe₃O₄@graphene is very low (no more than 20 emu/g) and the viscosity of paraffin is relatively large, the timely removal of Fe₃O₄@graphene in the melted paraffin may be somewhat difficult. The authors should provide the time for the hybrid NPs fully attracted to the side wall by a permanent magnet.
3. For comparison, the temperature rising curves of paraffin composites and the energy storage efficiency under OC process should be provided, since the authors claim that the MOC process can achieve higher storage efficiency than OC process.

Reviewer #2 (Remarks to the Author)

This is a well written, very interesting paper showing that magnetic tuning of optical absorbers can substantially enhance the charging rates of phase change materials (PCM). The main claim is that the proposed device can overcome the current limitations of thermally charging (slow thermal diffusion) and optically charging (opaqueness of the solid phase) PCMs. By removing the optically absorbing nanoparticles (Fe₃O₄-functionalized graphene) once the superficial sections of the PCM are melted, the subsequent layers of PCM can receive more light since the liquid PCM is more transparent than the solid phase.

Overall, the paper provides enough evidence to substantiate its main conclusions regarding device functionality and it is of interest to the scientific community. However, some of the information is questionable. Specifically, the authors should address the following comments:

- Novelty: Fe₃O₄ –graphene nanoparticles have been proposed before for very similar devices. For example, this recent publication might compromise its novelty: W. Wang et al, "Fe₃O₄-functionalized graphene nanosheet embedded phase change material composites: efficient magnetic- and sunlight-driven energy conversion and storage", J. Mater. Chem. A, 2017,5, 958-968. The approach there is focused more on the chemistry rather than on the device, but the idea is similar.
- The step by step charging mechanism is a model the authors have developed to match their experimental findings; it is not a novelty, it is rather a classical analytical approach solved by steps (Δx). It is a smart way to solve this problem, but it is far from novel. However, the authors fail to show how they determined the size of this step (Δx), how the solution varies with (Δx), and how the model was validated.
- Regarding the model and experiment comparison (Fig. 3d): why is the time interval shown on

the x-axis so short? The model seems to match well the OC experiments for $t > 50s$, but the charging times for this device are on the order of several minutes (300 s) and the last data points seem to indicate a deviation from the modeling results. What happens for $t > 70s$?

- Some Figures are missing key information: Fig. 4b: what are the charging times corresponding for these temperature profiles? Is Fig. 4.d. experiment or model results? How is the normalized stored heat (Fig4.c) calculated? What value for the latent heat of fusion is used as reference?
- Error bars are missing from the DSC data in Fig.5 e. Also, this data is hard to believe (the latent heat of fusion is increasing with particle loading?) and there is much published work to substantiate the opposite trend. When fillers are mixed with PCMs, the fillers do not undergo a phase change; therefore, the latent heat of the PCM+fillers should decrease with increasing particle loading. The authors should review their phase change experimental and integration procedures with the DSC.
- It is unclear what NP loading (%) is finally chosen for their solution: graphs are presented for 0.02 wt% in the first section as ideal (better charging times, indicating higher attenuation with larger particle wt%), but the solar thermal energy harvest section (L254) uses 0.1 wt%. A thorough analysis of the effect of particle loading on the device is missing.
- More information describing the magnetization procedure and its reversibility seems missing. In other words, the composites are shown to be stable after thermal cycling, but how about after optical + magnetically charging?
- Although the manuscript is overall well written, the order is strange. The DSC graphs and characterization of the optically charged composites (Fig. 5 and explanation) should be moved up to the beginning with the description of the functionalization of the graphene sheets (L112) or to the methods section (L340) or even to the supplementary information with the cycling graph (Sup. Fig 6). Additionally, the latent heat data of the composites is hard to believe and needs further analysis and explanation.
- Size of the device and up-scaling: it is clear that the proposed solution offers exciting options to increase the charging times of PCMs for solar harvesting/energy storage devices. It would be interesting to see how the authors plan to address the upscaling of their device from 0.4 cm x0.4 cm x2 cm. What are the current limitations in PCM thickness and particle loading for effectively tuning the distribution of optical absorbing particles?

This manuscript is very interesting and shows the potential of new type of devices combining optical and magnetic properties of nanoparticle-enhanced PCM. It is relatively new work, and the emphasis on demonstrating a device proposed here is certainly needed for the community to understand the behavior of such composites.

Resubmission after addressing the above mentioned comments is encouraged.

Reviewer #3 (Remarks to the Author)

Review of "Dynamic tuning of optical absorbers for accelerated solar-thermal energy storage"

This paper presents a novel optical charging approach of phase change materials (PCM). By incorporating optical absorbers in a PCM matrix and dynamic controlling their spatial using an external magnetic field, the incident radiation is preferentially absorbed near the phase change interface. Compared with conventional thermal charging approach, this strategy achieves higher charging rate as well as higher storage capacity as shown by both modeling and experimental results in the paper.

While the presented approach is novel, the practicability of this approach is not very clear. Specifically, we suggest the authors to provide more details about their visions as how this novel PCM solar absorber could be incorporated in current solar thermal systems. For example, solar thermal systems can have very different configurations from flat plate hot water heaters to high temperature concentrating solar power. While the experiments in this paper used concentrated solar flux, the phase change temperature of the PCM (60 oC) is rather low. This condition significantly differs from reality, where concentrating solar thermal systems typically operates at much higher temperature (~400 oC). We suggest the authors to clarify whether their approach is

applicable at higher temperature if their target application is power generation. On the other hand, if the target is water heating around 60 °C, the economy and efficiency of this approach needs to be justified against sensible heat storage using water. We also provide some detailed technical comments below. We believe that all of these points should be addressed before consideration of publication in Nature Communications.

List of technical comments:

1. The rationale behind the use of graphene and Fe₃O₄ nanoparticles is not clearly stated in the paper. What is the main advantage of this material choice compared to other materials? Is it cost effective?
2. Several temperature distribution profiles, such as Figure 2e,f, were shown in the paper. However, it is not clear how these temperature profiles are measured. The IR images that accompany these profiles all have color bars up to 50 °C, while the temperature profiles have maximum temperature over 100 °C.
3. (paragraph 1, page 4 and paragraph 2, page 8) "the converted thermal energy is accumulated at the conversion front and tends to be easily lost to the surrounding through radiation and natural convection.";

While the statement is true in general, we found this alone cannot possibly account for the observed difference in stored thermal energy for TC, OC and MOC as showed in Figure 4b, c. From Figure 4b, at position = 0.0 cm, the surface temperature T_s of TC, OC and MOC is 130 °C, 110 °C and 60 °C. For the worst case, we can estimate the heat loss to environment ($T_{amb} = 20$ °C) from a surface at 130 °C by convection and radiation assuming blackbody and natural convection:

$$q_{rad} = \sigma \epsilon (T_s^4 - T_{amb}^4) \approx 0.108 \text{ W/cm}^2, (\epsilon = 1)$$
$$q_{conv} = h(T_s - T_{amb}) \approx 0.110 \text{ W/cm}^2, (h = 10 \text{ W/(m}^2 \text{ K)})$$

Compared to the laser power density 4 W/cm² (40x sun), the total surface heat loss accounts for only 5.5% of the incident power. Thus, even by decreasing the surface heat loss to zero, it will only gain about 5.5% absolute efficiency. As a result, the claim of about ~118% and ~195% total stored thermal energy increase for OC and MOC modes compared to TC mode is likely to be due to other unclear reasons.

4. In Figure 3c, the simulated temperature distribution is much higher than the experiments (Figure 2e) for the TC modes. Some insights about the reasons as well as which result are closer to reality should be provided.
5. (paragraph 2, page 10) The solar thermal efficiency measurement procedure is very vague in both the main text and the supplemental material. As an important metrics and a major claim of the paper, we suggest the authors to elaborate more about their measurement so that the readers can better gauge the fidelity of their results.

We want to thank all the reviewers for their efforts in reviewing our manuscript and also for their valuable comments in helping improve the manuscript. Following is our point-to-point response to the reviewers' comments.

Reviewer #1 (Remarks to the Author):

The manuscript described dynamic tuning the distribution of Fe₃O₄@graphene in paraffin composites to achieve fast charging rates, large phase change enthalpy and high solar-thermal energy conversion efficiency. This research work is highly innovative, and will be of interest to others in the community. The manuscript can be published after the following questions are solved.

Reply: Thanks the reviewer for the encouraging comments and suggestions that help us improve the manuscript.

1. The characterization of Fe₃O₄@graphene needs to be supplemented and completed.

(1) The authors claim that the fabricated nanoparticles are magnetite but there is no clear evidence. XRD data themselves do not allow to make this conclusion because maghemite and magnetite are identical for XRD. The XPS spectrum of Fe₃O₄ needs to be supplemented.

Reply: Thanks the reviewer for the comment. In the revised manuscript, we included the XPS spectrum of Fe₃O₄@graphene hybrid nanoparticles (Figure S2), as suggested by the reviewer. Characteristic binding peaks corresponding to Fe²⁺ and Fe³⁺ were observed indicating the formation of magnetite. We also added the following statement in the manuscript (Page 3, last paragraph) to address the reviewer's concern.

"The XPS spectrum of Fe 2p level shows the existence of both Fe²⁺ and Fe³⁺ binding states in the hybrid NPs indicating the formation of Fe₃O₄NPs⁴⁰ (Supplementary Fig. S2b)."

(2) The authors should provide more evidence for the reduction of GO into graphene. For example, the XPS spectrum of C 1s for GO and Fe₃O₄@graphene.

Reply: Thanks the reviewer for the comment. In the revised manuscript, we included the XPS spectrum of GO and Fe₃O₄@graphene hybrid NPs (Figure S2), as suggested by the reviewer. From the spectrum of C 1s level, we found that the peak area of C-O and C=O decreased after the reaction, indicating the reduction of GO. We added the following description in the manuscript (Page 4, 1st paragraph) to address the reviewer's concern.

"The XPS spectrum of C 1s level also shows the decrease of C-O and C=O binding peak intensity from GO to Fe₃O₄@graphene NPs (Supplementary Fig. S2c and S2d)."

(3) M-H curve measured from -30 kOe to 30 kOe cannot determine the superparamagnetic behavior. The magnified hysteresis curve or ZFC/FC curve may be adopted to characterize the magnetic behavior. The authors can refer to the following literatures: Journal of Materials

Chemistry A 2017, 5, 958-968.

Reply: Thanks the reviewer for the suggestion. In the revised manuscript, we provided the magnified hysteresis curve from -5 kOe to 5 kOe (Figure S5a), as suggested by the reviewer. We also measured temperature-dependent magnetization behavior of Fe₃O₄@graphene hybrid NPs by using zero-field cooling (ZFC) and field-cooling (FC) method (Figure S5b). The ZFC curve shows a superparamagnetic blocking temperature of 110 K. We added the following statement in the manuscript (Page 8, last paragraph) and cited the ref. 40 (J. Mater. Chem. A 2017, 5, 958-968) and Ref. 49 to address the reviewer's concern.

"The zero-field cooling (ZFC) and field-cooling (FC) magnetization curves indicate that the Fe₃O₄@graphene NPs have a superparamagnetic blocking temperature of 110 K that is associated with the superparamagnetic Fe₃O₄ NPs^{40, 49} (Supplementary Fig. S5)."

2. Since the saturation magnetization of Fe₃O₄@graphene is very low (no more than 20 emu/g) and the viscosity of paraffin is relatively large, the timely removal of Fe₃O₄@graphene in the melted paraffin may be somewhat difficult. The authors should provide the time for the hybrid NPs fully attracted to the side wall by a permanent magnet.

Reply: Thanks the reviewer for the comment. For the sample contained within the cuvette (0.4 cm x 0.4 cm x 2 cm), it was observed that Fe₃O₄@graphene NPs in melted paraffin were fully attracted by a magnet with a magnetic field strength of ~0.3 Tesla to the side wall within ~6 seconds. We included this information in the following statement in the revised manuscript (Page 8, last paragraph) to address the reviewer's concern.

"Once the PCM composites are melted, within ~6 s the hybrid NPs are fully attracted to the side wall by a permanent magnet with a magnetic field strength of ~0.3 Tesla."

3. For comparison, the temperature rising curves of paraffin composites and the energy storage efficiency under OC process should be provided, since the authors claim that the MOC process can achieve higher storage efficiency than OC process.

Reply: Thanks the reviewer for the suggestion. Both the temperature rising curves and the energy storage efficiencies for both the OC process and the MOC process are included in the revised Supplementary Information (Fig. S9; Page S3, 2nd paragraph), as suggested by the reviewer. This evaluation approach based on temperature rising curve is used by many others reported efforts (Ref. 24, 25, 27, 33, 35, 37, 40) in characterizing the solar-thermal energy storage efficiency during the phase change process. Our intention of using this approach in this work is to compare the performance of our PCM composite system with the performance of other similar PCM composite systems using the same characterization approach. Our result did show that the efficiency during the MOC charging process (~92.4%) is much higher than the reported efficiency of 40%-60% in paraffin CNT sponge systems (Ref. 24) and is comparable to the highest value

reported so far but with much lower loading concentrations. In the original manuscript, we only mentioned the efficiency of our PCM composite during OC process. In the revised manuscript, we moved the OC data to the Supplementary Information to compare it with the MOC data. We revised the first 3 sentences of the 2nd paragraph, page 11 to make clear our intention and also to highlight the benefits of using the MOC process:

“To compare the charging performance of the PCM composite systems developed in this work to other PCM composite systems, we measured the temperature rising curves of thin samples (with a thickness ~ 3 mm). Under MOC charging mode, the composite sample with 0.1 wt% NPs has shown a high energy storage efficiency of 92.4% (Fig. S9), which is much higher than the reported efficiency of 40%-60% in paraffin CNT sponge systems²⁴ and is comparable to the highest value reported so far but with much lower loading concentrations^{25, 27, 33, 35, 37, 40}.”

Reviewer #2 (Remarks to the Author):

This is a well written, very interesting paper showing that magnetic tuning of optical absorbers can substantially enhance the charging rates of phase change materials (PCM). The main claim is that the proposed device can overcome the current limitations of thermally charging (slow thermal diffusion) and optically charging (opaqueness of the solid phase) PCMs. By removing the optically absorbing nanoparticles (Fe₃O₄-functionalized graphene) once the superficial sections of the PCM are melted, the subsequent layers of PCM can receive more light since the liquid PCM is more transparent than the solid phase.

Overall, the paper provides enough evidence to substantiate its main conclusions regarding device functionality and it is of interest to the scientific community. However, some of the information is questionable. Specifically, the authors should address the following comments:

1. Novelty: Fe₃O₄ –graphene nanoparticles have been proposed before for very similar devices. For example, this recent publication might compromise its novelty: W. Wang et al, “Fe₃O₄-functionalized graphene nanosheet embedded phase change material composites: efficient magnetic- and sunlight-driven energy conversion and storage”, J. Mater. Chem. A, 2017,5, 958-968. The approach there is focused more on the chemistry rather than on the device, but the idea is similar.

Reply: Thanks the reviewer for the comments and bringing to our attention this reference by Professor Tang (J. Mater. Chem. A, 2017, 5, 958-968; we cited this reference as ref. 40 in the revised manuscript). Professor Tang’s work demonstrated a novel approach in enabling dual-mode charging process that involves both magnetic charging process and optical charging process. The core concept of this work, however, is different from Professor Tang’s work in the following two aspects:

a) Professor Tang’s team used superparamagnetic Fe₃O₄ NPs as magnetic-to-thermal energy convertors during the magnetic charging process of their PCM using alternative magnetic field – there is no movement of photon absorbers involved. In this work, we are not using the superparamagnetic Fe₃O₄ NPs as magnetic-to-thermal energy convertors because there

is no magnetic charging process involved in this study, and we did not convert magnetic energy into thermal energy. We used superparamagnetic Fe_3O_4 NPs to move the photon absorbers (graphene nanosheets) to the side wall in a static magnetic field during the optical charging process.

- b) In Professor Tang's work, the optical charging still relies on the improved thermal conductivity of the PCM composites (page 11 of the reference). In this work, the enhanced charging rate does not rely on the thermal conduction within the composites. The charging rate enhancement came from the rapid transportation of photons through the timely removal of photon absorbers in the melted PCM.

2. The step by step charging mechanism is a model the authors have developed to match their experimental findings; it is not a novelty, it is rather a classical analytical approach solved by steps (Δx). It is a smart way to solve this problem, but it is far from novel. However, the authors fail to show how they determined the size of this step (Δx), how the solution varies with (Δx), and how the model was validated.

Reply: Thanks the reviewer for the comment and stimulating questions. It is indeed challenging to determine the size of Δx , as we discussed in the revised Supplementary Information (Page S6, 3rd paragraph). In general, the smaller Δx , the better in modeling the charging process. In Fig. S3a, we showed that by varying Δx , there is a change with the position plots. As Δx gets smaller, the change between the position plots gets smaller. When Δx is $\sim 0.01\text{cm}$, there is not much change and thus we used $\Delta x = 0.01\text{ cm}$ in the modeling. In the revised manuscript, we added the following statements in the manuscript (Page 7, 1st paragraph) and in the Supplementary Information (Page S6, 3rd paragraph) to address the reviewer's concern.

Statement added in the manuscript (Page 7, 1st paragraph):

"To simulate OC process, appropriate selection of step size is critical. By trying with different step sizes, it was found that when Δx approaches 0.01 cm the simulation results converge and further decreasing Δx does not change much the simulated interface movement (Fig. S3a)."

Statement added in the Supplementary Information (Page S6, 3rd paragraph):

"To determine the appropriate step size we tried different Δx . The results show that when Δx approaches 0.01 cm, the simulated propagation curves of charging interface converge (Fig. S3a). Based on Fig S3a, a step size of 0.01 cm was used in our model to simulate the charging process with appropriate amount of computation effort."

3. Regarding the model and experiment comparison (Fig. 3d): why is the time interval shown on the x-axis so short? The model seems to match well the OC experiments for $t > 50\text{s}$, but the charging times for this device are on the order of several minutes (300 s) and the last data points seem to indicate a deviation from the modeling results. What happens for $t > 70\text{s}$?

Reply: Thanks the reviewer for the comments. We added Figure S3b to show the comparison

between the modeling result and the experiment result over time longer than 70s to address the reviewer's concern. For $t > 50$ s, the difference between the modeling and the experimental results became larger due to the increase of heat loss as the temperature of the composites increases. With the classical analytic model we used in the simulation of the charging process, it is challenging to consider the heat loss (Ref. 30, Ref. 47-48). Without considering heat loss in classical analytic models the simulation results tend to overestimate the charging rate. Similar treatment and overestimation of temperature/charging rate have been reported by Li et al. (Ref. 30, *Energ. Environ. Sci.* 7, 1185-1192, 2014), Shidfar et al. (Ref. 47, *Opt. Laser Technol.* 41, 280-284, 2009) and Shen et al. (Ref. 48, *Opt. Laser Technol.* 33, 533-537, 2001) in their work, respectively. In the revised manuscript, we added Figure S3b and the following statements in the manuscript (Page 7, 1st paragraph) to address the reviewer's concern:

"As shown by Fig. 3d, the simulation results confirm the much faster charging rates in the OC mode and match well with experimental data within the initial 60 s. With prolonged charging time, the theoretical model tends to overestimate the movement of charging interface and temperature due to the increase of heat loss as the temperature of the composites increases.^{30, 47, 48} (Fig. S3b)."

4. Some Figures are missing key information: Fig. 4b: what are the charging times corresponding for these temperature profiles? Is Fig. 4.d. experiment or model results? How is the normalized stored heat (Fig4.c) calculated? What value for the latent heat of fusion is used as reference?

Reply: Thanks the reviewer for the comments. The charging time for the temperature profiles in Fig. 4b is 270 s. Fig. 4d shows the experimental results. The normalized stored heat in Fig. 4c was calculated by comparing the amount of heat stored within PCM under different charging mode and by using MOC mode as the benchmark. For a specific charging mode, we calculated sensible heat ($C_p \times m \times \Delta T$) and latent heat ($m \times \Delta H$), where m is mass, C_p is heat capacity (for solid: 2.4 J/g K, for liquid: 2.8 J/g K), and ΔH is heat of fusion. Based on the final temperature distribution profile in Fig. 4b, the amount of stored sensible heat within solid PCM and sensible heat stored within liquid PCM were calculated through integrating the temperature profile along the charging distance. The latent heat of fusion used is 174 J/g. In the revised manuscript, we added Figure S6 and the following statement in the Supplementary Information (page S2, last paragraph) to address the reviewer's concern:

"Based on the final temperature distribution profiles, the amount of stored solar-thermal energy including the sensible heat stored within solid PCM, the sensible heat stored within liquid PCM, and the latent heat were calculated. As shown by Fig. S6, for example in OC mode in Fig. 4b, the stored sensible heat was calculated through integrating the temperature profile along the charging distance by using $H_{sensible} = \int_{x_1}^{x_2} c_p \times T \times \rho \times A \times dx$, where ρ is the density of PCM, c_p is the heat capacity (liquid: 2.8 J/g K, solid: 2.4 J/g K), A is the cross-sectional area of the sample, x is the sample length along the charging direction. In Fig. 4b, for the sensible heat stored within liquid PCM under OC mode, x_1 and x_2 are 0 and 1.3 cm, respectively. For the sensible heat stored

within solid PCM, x_1 and x_2 are 1.3 and 2 cm, respectively. The latent heat stored was estimated by using the mass in liquid phase and the latent heat of fusion: $H_{latent} = M_{liquid} \times H_{fusion} = \rho \times A \times (x_2 - x_1) \times H_{fusion}$, where $(x_2 - x_1)$ stands for the melted length of the sample. The heat of fusion (H_{fusion}) for PCM composites with 0.02 wt% of Fe_3O_4 @graphene hybrid NPs was 174 J/g based on the DSC measurement. The same approach was used to calculate the stored solar-thermal energy under TC and MOC mode. The relative amount of stored energy was compared among different charging modes."

5. Error bars are missing from the DSC data in Fig.5 e. Also, this data is hard to believe (the latent heat of fusion is increasing with particle loading?) and there is much published work to substantiate the opposite trend. When fillers are mixed with PCMs, the fillers do not undergo a phase change; therefore, the latent heat of the PCM+fillers should decrease with increasing particle loading. The authors should review their phase change experimental and integration procedures with the DSC.

Reply: Thanks the reviewer for the comments and suggestions. We further reviewed our DSC measurement and analysis processes, repeated the measurements and added error bars in the DSC data in Fig. 2h in the revised manuscript (Fig. 5e in the original manuscript), as suggested by the reviewer. We agree that in many cases that the latent heat of fusion decreases when PCMs are mixed with fillers. In the case of carbon-based fillers, however, there are also many reports (Ref. 24, 41-46) of increased latent heat for PCMs mixed with carbon-based fillers due to the interaction between the PCM molecules and the surface of carbon fillers (for example, Ref 24: ACS Nano: **6**, 10884-10892 (2012); Ref. 43: Sol. Energy Mater. Solar Cells, **95**, 1811-1818 (2011); Ref. 44: Adv. Mater. **25**, 2554-2560 (2013)). We added those references in the revised manuscript and revised the following statement to address the reviewer's concern (Page 6, 1st paragraph) :

"Similar to the other PCM systems doped with carbon-based fillers^{24, 41-46}, the strong intermolecular interaction that favors the alignment of alkane chains at the paraffin/graphene interface might help improve the crystallinity that leads to the increase of phase change enthalpy of paraffin wax composites. The OLA ligands that are attached to the surface of graphene might also help enhance such intermolecular interaction in improving the crystallinity of the PCMs."

6. It is unclear what NP loading (%) is finally chosen for their solution: graphs are presented for 0.02 wt% in the first section as ideal (better charging times, indicating higher attenuation with larger particle wt%), but the solar thermal energy harvest section (L254) uses 0.1 wt%. A thorough analysis of the effect of particle loading on the device is missing.

Reply: Thanks the reviewer for comment. In the first section, we started the investigation from the OC mode (Figure 2 and Figure 3). Under normal OC mode 0.02 wt% is used because its attenuation is less than particle loading of higher wt%, as commented by the reviewer as well. We still used 0.02 wt% in Figure 4 to compare the performance of OC and MOC. For the solar-thermal energy harvest section (Figure 5) we used a different NP loading (0.1 wt%) to show

that the advantage of using MOC is applicable to a range of concentrations. In the revised manuscript, we added the following discussion of particle loading effect in the discussion section (Page 11, 2nd paragraph) to address the reviewer's concern:

"Effective tuning of the distribution of photothermal converters within melted PCM is the key to realizing accelerated charging process. During solar charging experiment, it was found that the Fe₃O₄@graphene NP loading up to 0.1 wt% could be effectively attracted to the side wall of a petri dish with a diameter of ~ 4 cm. Loading higher than 0.1 wt% leads to lowering the heat of fusion after the initial enhancement of heat fusion with loading <0.1 wt%. Further increasing NP loading would also increase the viscosity of the melted PCM composites^{43, 51}, which would limit the movement of the hybrid NPs. Developing hybrid photothermal converters with stronger magnetic response and applying stronger magnets such as electromagnets would enable effective manipulation of the absorbers within large size charging systems."

7. More information describing the magnetization procedure and its reversibility seems missing. In other words, the composites are shown to be stable after thermal cycling, but how about after optical + magnetically charging?

Reply: Thanks the reviewer for the comment. Following the reviewer's suggestion, we carried out repeated charging tests under MOC mode for 100 cycles and measured the DSC curves. As shown by Figure S10b (Supplementary Information) in the revised manuscript, the DSC curves are almost overlapped indicating excellent stability of the PCM composites under repeated magnetically-enhanced optical charging.

8. Although the manuscript is overall well written, the order is strange. The DSC graphs and characterization of the optically charged composites (Fig. 5 and explanation) should be moved up to the beginning with the description of the functionalization of the graphene sheets (L112) or to the methods section (L340) or even to the supplementary information with the cycling graph (Sup. Fig 6). Additionally, the latent heat data of the composites is hard to believe and needs further analysis and explanation.

Reply: Thanks the reviewer for the suggestions and comments. As suggested by the reviewer, we moved the DSC graphs from Figure 5 in the original manuscript to Figure 2 (Figure 2g and 2h) in the revised manuscript. The corresponding explanations were also moved to the beginning section of the manuscript.

Also thanks again for the reviewer's comments on the latent heat data. As we explained in the response to the question 5 from the reviewer, in the case of carbon-based fillers, there are also many reports (Ref. 24, 41-46) of increased latent heat for PCMs mixed with carbon-based fillers due to the interaction between the PCM molecules and the surface of carbon fillers (for example, Ref 24: ACS Nano: 6, 10884-10892 (2012); Ref. 43: Sol. Energy Mater. Solar Cells, 95, 1811-1818 (2011); Ref. 44: Adv. Mater. 25, 2554-2560 (2013)). We added those references in the revised

manuscript and revised the following statement to address the reviewer's concern (Page 6, 1st paragraph).

“Similar to the other PCM systems doped with carbon-based fillers^{24, 41-47}, the strong intermolecular interaction that favors the alignment of alkane chains at the paraffin/graphene interface might help improve the crystallinity that leads to the increase of phase change enthalpy of paraffin wax composites. The OLA ligands that are attached to the surface of graphene might also help enhance such intermolecular interaction in improving the crystallinity of the PCMs.”

9. Size of the device and up-scaling: it is clear that the proposed solution offers exciting options to increase the charging times of PCMs for solar harvesting/energy storage devices. It would be interesting to see how the authors plan to address the upscaling of their device from 0.4 cm x0.4 cm x2 cm. What are the current limitations in PCM thickness and particle loading for effectively tuning the distribution of optical absorbing particles?

Reply: Thanks the reviewer for the comment. The upscaling will depend on whether we can still effectively tune the distribution of optical absorbing particles at large scale. The effective tuning of the distribution of optical absorbing particles depends on the magnetic force applied to the Fe₃O₄@graphene NPs. The thickness of the PCM will affect the magnetic force because the magnetic field strength decreases with the increase of distance from the surface of magnets. During our solar energy conversion experiment, we placed our PCM composites (0.01-0.1 wt%) within a petri dish (4 cm in diameter) and found that the common magnet with a field strength of ~ 0.3 Tesla could effectively attract the hybrid particles to the side wall. By using hybrid photothermal converters with stronger magnetic response and applying stronger magnetic field generated by electromagnets the upscaling of the system is possible. As to particle loading concentration, high loading leads to lowering the heat of fusion after the initial enhancement of heat fusion with loading <0.1 wt%. Further increasing NP loading would also increase the viscosity of the melted PCM composites that limits the movement of the hybrid NPs.

In the revised manuscript, we provided the following discussion on scaling up the device, thickness and particle loading effect on magnetic manipulation in the discussion section (Page 11, 2nd paragraph).

“Effective tuning of the distribution of photothermal converters within melted PCM is the key to realizing accelerated charging process. During solar charging experiment, it was found that the Fe₃O₄@graphene NP loading up to 0.1 wt% could be effectively attracted to the side wall of a petri dish with a diameter of ~ 4 cm. Loading higher than 0.1 wt% leads to lowering the heat of fusion after the initial enhancement of heat fusion with loading <0.1 wt%. Further increasing NP loading would also increase the viscosity of the melted PCM composites^{43, 51}, which would limit the movement of the hybrid NPs. Developing hybrid photothermal converters with stronger magnetic response and applying stronger magnets such as electromagnets would enable effective manipulation of the absorbers within large size charging systems. In addition to the low-temperature applications by using paraffin as the PCM, the MOC approach could also be

applicable for rapid high-temperature solar-thermal energy charging with the development of high-temperature dual-functional solar-thermal converters such as low-cost carbon-based absorber and using molten salt as the PCM matrix. The demonstrated advantageous features including extremely low loading requirement, effective tuning distribution with external magnets, and facile preparation process hold the promise of scaling up the charging system for practical solar-thermal applications."

10. This manuscript is very interesting and shows the potential of new type of devices combining optical and magnetic properties of nanoparticle-enhanced PCM. It is relatively new work, and the emphasis on demonstrating a device proposed here is certainly needed for the community to understand the behavior of such composites. Resubmission after addressing the above mentioned comments is encouraged.

Reply: Thanks the reviewer again for the encouraging comments and suggestions in improving the manuscript.

Reviewer #3 (Remarks to the Author):

Review of "Dynamic tuning of optical absorbers for accelerated solar-thermal energy storage"

This paper presents a novel optical charging approach of phase change materials (PCM). By incorporating optical absorbers in a PCM matrix and dynamic controlling their spatial using an external magnetic field, the incident radiation is preferentially absorbed near the phase change interface. Compared with conventional thermal charging approach, this strategy achieves higher charging rate as well as higher storage capacity as shown by both modeling and experimental results in the paper.

While the presented approach is novel, the practicability of this approach is not very clear. Specifically, we suggest the authors to provide more details about their visions as how this novel PCM solar absorber could be incorporated in current solar thermal systems. For example, solar thermal systems can have very different configurations from flat plate hot water heaters to high temperature concentrating solar power. While the experiments in this paper used concentrated solar flux, the phase change temperature of the PCM (60 °C) is rather low. This condition significantly differs from reality, where concentrating solar thermal systems typically operates at much higher temperature (~400 °C). We suggest the authors to clarify whether their approach is applicable at higher temperature if their target application is power generation. On the other hand, if the target is water heating around 60 °C, the economy and efficiency of this approach needs to be justified against sensible heat storage using water. We also provide some detailed technical comments below. We believe that all of these points should be addressed before consideration of publication in Nature Communications.

Reply: Thanks the reviewer for the comments. As we stated in the original manuscript, we used

the low temperature PCM as a model system to demonstrate the enhancement of charging rate through the dynamic tuning of the photon absorbers. We agree with the reviewer that it will be challenging for this strategy to compete with sensible heat storage using water for water heating at relatively low temperature (for example, 60 °C). In fact, most of PCM-based systems will not be able to compete with sensible heat storage using water in those applications. At these low temperatures, potential applications for our PCM systems might be those that potentially benefit from the use of PCM, including thermal storage in buildings, thermoregulation, space power management and solar drying in food industry (Ref. 4 and Ref. 17-19).

Even though we demonstrated the magnetically-enhanced optical charging strategy with the low temperature PCM system, such strategy can also be applicable for the high-temperature PCMs. The key for the high-temperature application of this strategy is to develop the stable high-temperature dual-functional solar-thermal converting materials that can be dynamically tuned, which currently are under investigation in our lab. Our preliminary results with some other carbon-based converters showed that these converters are stable during the thermal charging of molten salt (with melting point of ~220 °C). The charging of such system in OC mode showed much faster charging rates than in convention TC mode (see figure below). We are currently investigating the possible magnetic particles that can be stably attached to these carbon-based converters at high temperature to enable the MOC mode at high temperature. In the revised manuscript, we included the following statement on scalability in the discussion section (Page 11, 2nd paragraph):

“Developing hybrid photothermal converters with stronger magnetic response and applying stronger magnets such as electromagnets would enable effective manipulation of the absorbers within large size charging systems. In addition to the low-temperature applications by using paraffin as the PCM, the MOC approach could also be applicable for rapid high-temperature solar-thermal energy charging with the development of high-temperature dual-functional solar-thermal converters such as low-cost carbon-based absorber and using molten salt as the PCM matrix. The demonstrated advantageous features including extremely low loading requirement, effective tuning distribution with external magnets, and facile preparation process hold the promise of scaling up the charging system for practical solar-thermal applications.”

Figure 1. Comparison between thermal charging and optical charging of molten salt composite. **a**, SEM image of the carbon-based nanofillers dispersed in eutectic salts. **b**, Comparison of infrared images of thermally charged eutectic salts, thermally charged eutectic salt composites filled with carbon-based nanofillers, and optically charged composites after illumination for 10 s

(from left to right). **c**, Temperature distribution profile along the charging direction. The sample was directly illuminated by a green laser beam (532 nm) in optical charging process and illuminated onto a black aluminum absorber in direct contact with eutectic salt in thermal charging process, respectively.

List of technical comments:

1. The rationale behind the use of graphene and Fe₃O₄ nanoparticles is not clearly stated in the paper. What is the main advantage of this material choice compared to other materials? Is it cost effective?

Reply: Thanks the reviewer for the comment. In our work, the Fe₃O₄@graphene hybrid NPs combine the good solar-thermal conversion capability from graphene and magnetic response from the Fe₃O₄ NPs, thus allowing for accelerating the optical charging through magnetically tuning the distribution of the photothermal converters within the PCM. Graphene has been considered as one of effective solar-thermal converting materials, and Fe₃O₄ NPs are widely used as one of the cost-effective magnetic materials. Additionally, the Fe₃O₄@graphene NPs have been surface-capped so they can be well dispersed within organic PCM. Based on these considerations, we chose Fe₃O₄@graphene hybrid NPs as the model dual-functional fillers in our system. In the revised manuscript, we added the following statement to address the reviewer's concern.

Page 3, 1st paragraph;

"The Fe₃O₄@graphene hybrid NPs were used due to the good optical absorption of graphene and also the good magnetic-response of Fe₃O₄ NPs."

Page 11, 1st paragraph:

"The good dispersion enables the effective solar absorption with the extremely low loading concentration of the absorbers."

2. Several temperature distribution profiles, such as Figure 2e,f, were shown in the paper. However, it is not clear how these temperature profiles are measured. The IR images that accompany these profiles all have color bars up to 50 °C, while the temperature profiles have maximum temperature over 100 °C.

Reply: Thanks the reviewer for the comments. The temperature profiles were acquired by drawing a line along the center of the charged sample in the recorded IR images. Our initial intention for choosing the same temperature scale bars up to 50 °C is to visually identify the melted region from the IR image as the melting temperature of paraffin is around 50 °C. In the revised manuscript, we revised the scale bars and added the following statement in Fig 2 caption (Page 4) to address the reviewer's concern.

"The temperature profiles were extracted from the central line of the IR images."

3. (paragraph 1, page 4 and paragraph 2, page 8) “the converted thermal energy is accumulated at the conversion front and tends to be easily lost to the surrounding through radiation and natural convection.”;

While the statement is true in general, we found this alone cannot possibly account for the observed difference in stored thermal energy for TC, OC and MOC as showed in Figure 4b, c. From Figure 4b, at position = 0.0 cm, the surface temperature T_s of TC, OC and MOC is 130 °C, 110 °C and 60 °C. For the worst case, we can estimate the heat loss to environment ($T_{amb} = 20$ °C) from a surface at 130 °C by convection and radiation assuming blackbody and natural convection:

$$q_{rad} = \sigma \epsilon (T_s^4 - T_{amb}^4) \approx 0.108 \text{ W/cm}^2, (\epsilon = 1)$$

$$q_{conv} = h(T_s - T_{amb}) \approx 0.110 \text{ W/cm}^2, (h = 10 \text{ W/(m}^2 \text{ K)})$$

Compared to the laser power density 4 W/cm² (40x sun), the total surface heat loss accounts for only 5.5% of the incident power. Thus, even by decreasing the surface heat loss to zero, it will only gain about 5.5% absolute efficiency. As a result, the claim of about ~118% and ~195% total stored thermal energy increase for OC and MOC modes compared to TC mode is likely to be due to other unclear reasons.

Reply: Thanks the reviewer for the careful reading of our manuscript. Our experimental setup includes a quartz container that is transparent to the charging light and can sustain the high temperature experienced by the PCM during the TC and OC charging processes. The container also had an open top so we can record the temperature profile using an IR camera during the various charging processes. With such setup, besides the radiation and convection heat losses at the front (as mentioned by the reviewer), there are also radiation and convection heat losses at the top surface and the conduction heat loss to the container. The difference in the charging mechanisms led to the difference in the temperature profiles, which also resulted in the differences in the temperature-dependent heat losses. Such temperature-dependent heat losses all contributed to the differences in the energy stored in different charging processes. In the revised manuscript, we added the following statement (Page 9, 1st paragraph) to address the reviewer’s concern:

“The difference in the charging mechanisms led to the difference in the temperature profiles, which also resulted in the differences in the temperature dependent heat losses that include the radiation and convection heat losses at the front and top surfaces, and also the conduction heat losses to the side walls of the container.”

4. In Figure 3c, the simulated temperature distribution is much higher than the experiments (Figure 2e) for the TC modes. Some insights about the reasons as well as which result are closer to reality should be provided.

Reply: Thanks the reviewer for the comment and suggestion. Considering the heat loss, the experimental results are closer to reality than the simulation results. With the classical analytic

model we used in the simulation of the charging process, it is challenging to consider the heat loss (Ref. 30, Ref. 47-48). Without considering heat loss in classical analytic models the simulation results tend to overestimate the temperature. Similar treatment and overestimation of temperature/charging rate have been reported by Li et al. (Ref. 30, *Energ. Environ. Sci.* 7, 1185-1192, 2014), Shidfar et al. (Ref. 47, *Opt. Laser Technol.* 41, 280-284, 2009) and Shen et al. (Ref. 48, *Opt. Laser Technol.* 33, 533-537, 2001) in their work, respectively. In the revised manuscript, we added the following statements in the manuscript (Page 7, 1st paragraph) to address the reviewer's concern:

“With prolonged charging time, the theoretical model tends to overestimate the movement of charging interface and temperature due to the increase of heat loss as the temperature of the composites increases.”^{30, 47, 48,}

5. (paragraph 2, page 10) The solar thermal efficiency measurement procedure is very vague in both the main text and the supplemental material. As an important metrics and a major claim of the paper, we suggest the authors to elaborate more about their measurement so that the readers can better gauge the fidelity of their results.

Reply: Thanks the reviewer for the comment. We included the following detailed procedure in the Supplementary Information (Page S3, 2nd paragraph) to address the reviewer's concern. This approach of using temperature rising curve is used by many others reported efforts (Ref. 24, 25, 27, 33, 35, 37, 40) in characterizing the solar-thermal energy storage efficiency during the phase change process.

“As shown by Fig. S8a, thin composite samples with thicknesses of ~3 mm were first placed in a transparent plastic petri dish (with a diameter of ~4 cm) that was thermally insulated with polystyrene foam and illuminated by simulated solar light under a power density of 0.3 W/cm². The temperature evolution was monitored by the thermocouple, which was inserted into the PCM and recorded by a data acquisition system (Agilent 34972A, Agilent Technologies, USA). Thermal energy storage efficiency during solar charging was calculated based on the ratio of the stored thermal energy in paraffin wax to the optical illumination energy received by the composite material over the phase change process. The efficiency can be described by the following formula:

$$\eta = \frac{m \Delta H}{Pt} \quad (S1)$$

where m is the weight of the charged sample, ΔH is heat of fusion of the optically charged composite samples that is determined by DSC measurement; P is the solar power that can be calculated by multiplying solar power density with the area of the sample; t is the charging time during the phase change process. The phase-change period (t) was determined by the tangential method through the analysis of the heating curve²⁻⁸ and was defined as the time difference between the starting point and the end point of the phase change process.”

Reviewers' Comments:

Reviewer #1:

Remarks to the Author:

This paper has been modified, and the corresponding data has been supplemented. In view of the originality of the article, I recommend this article to be published in Nature Communications.

Reviewer #2:

Remarks to the Author:

Although most questions were addressed adequately in the rebuttal, I am afraid the authors misunderstood my comments regarding the latent heat data. As the authors acknowledge in the rebuttal (but unfortunately not in the revised text) it is much more common to find a decrease in latent heat of composite PCM when nanoparticles are added (as opposed to an increase as the authors show here). The authors have found certain publications which show the opposite trend (increase in latent heat by adding particles); however these "anomalous" results have not been fully explained yet or accepted by the community. This trend is "rarely" found in the literature (as Ref. 44 states). Ref. 43 (which is incorrectly cited) also explains in its text that the changes in phase change enthalpy are negligible at low particle loading and the latent heat decreases with high particle concentrations. We have tested other composite PCM and have found similar results. I am not comfortable with the authors stating a hypothetical, unverified explanation to their latent heat enhancements as an accepted mechanism and I urge them to at least mention the controversy regarding these results.

I realize these are not the main results or the central contribution of this work, but it is unfortunate that doubtful results are included in a manuscript that seems otherwise relatively sound.

I am also not sure Reviewer #3's comments in point 3) have been appropriately addressed.

Reviewer #3:

Remarks to the Author:

We think the authors have done a good job addressing most of the comments in the previous round of review. The added discussion on the potential application in high-temperature solar thermal conversion process justifies the implication of the presented low-temperature demonstration. The added details clarify the experimental procedures and greatly help readers to interpret the results. Overall, we think the manuscript has improved a lot. Meanwhile, we have one additional suggestion regarding the heat transfer analysis for the authors to consider. After making the heat transfer analysis more rigorous, we think the manuscript is ready to be published in *Nature Communications*.

Suggestion on heat transfer analysis:

The suggestion is related to comment 3 and 4 in the previous review. In comment 3, we pointed out that the reduced front surface heat loss of MOC compared to TC and OC is not enough to explain the observed improved charging efficiency. In the response, the discrepancy is explained by the difference in temperature profiles, which also lead to different temperature dependent heat losses from both the front and top surfaces. In general, we agree with the authors that the improved charging efficiency must relate to the difference in heat losses, which are ultimately determined by the temperature profiles. However, we think the exact reason that MOC performs better than TC and OC is not clearly stated from a heat transfer perspective. The added sentence in the manuscript states that the temperature profile and heat losses are different among the three charging modes. However, the reason that MOC has the minimum heat loss is not clear. From Figure 4b, MOC has the most uniform and broad temperature profile. Graphically, since heat losses are represented by the area between the measured temperature profile and the ambient temperature, it is not immediately obvious that MOC has significantly reduced heat loss compared to the other two modes. We believe the reason that MOC has the minimum heat loss is of the central importance of this work. So we suggest the authors elaborate more here.

Another related suggestion is about neglecting heat losses in the theoretical model. We understand that the actual heat loss is hard to track by the simple analytical approach. However, simple estimation can usually be used to gain useful understandings about the model. For example, as in the energy conservation in Eq. S15, an additional heat loss term can be added to the left-hand side of the equation:

$$q_{loss} = h_{loss} P \Delta x (T_i - T_{amb}) / A_{cuvette}$$

where h_{loss} is the heat loss coefficient ($W/m^2 K$), P is the perimeter of the cuvette (m), $A_{cuvette}$ is the cuvette cross section area (m^2), T_{amb} is the ambient temperature. While the actual h_{loss} is challenging to quantify, some guesses can be made based on the insulation material property/geometry, convection correlation, and radiation. With the help of the improved model, the simulation should be closer to the experimental results (Figure 3c). In addition, the model should also shed light on the reason why MOC has better performance compared to the other modes.

We sincerely thank all the reviewers for their valuable comments and insightful suggestions. Based on the reviewers' comments and suggestions, we further revised our manuscript to address the reviewers' concerns. Following is our point-to-point response to the reviewers' comments.

Reviewer #1 (Remarks to the Author):

This paper has been modified, and the corresponding data has been supplemented. In view of the originality of the article, I recommend this article to be published in Nature Communications.

Reply: Many thanks again for the comments and suggestions that helped us improve the manuscript.

Reviewer #2 (Remarks to the Author):

Although most questions were addressed adequately in the rebuttal, I am afraid the authors misunderstood my comments regarding the latent heat data. As the authors acknowledge in the rebuttal (but unfortunately not in the revised text) it is much more common to find a decrease in latent heat of composite PCM when nanoparticles are added (as opposed to an increase as the authors show here). The authors have found certain publications which show the opposite trend (increase in latent heat by adding particles); however these "anomalous" results have not been fully explained yet or accepted by the community. This trend is "rarely" found in the literature (as Ref. 44 states). Ref. 43 (which is incorrectly cited) also explains in its text that the changes in phase change enthalpy are negligible at low particle loading and the latent heat decreases with high particle concentrations. We have tested other composite PCM and have found similar results. I am not comfortable with the authors stating a hypothetical, unverified explanation to their latent heat enhancements as an accepted mechanism and I urge them to at least mention the controversy regarding these results.

I realize these are not the main results or the central contribution of this work, but it is unfortunate that doubtful results are included in a manuscript that seems otherwise relatively sound. I am also not sure Reviewer #3's comments in point 3) have been appropriately addressed.

Reply: Many thanks for the comments and suggestions related to the latent heat. We agree with the reviewer that the decrease of the latent heat is more common than the increase of the latent heat as the particle loading increases, and the exact mechanism for the increase of the latent heat is still under debate. As suggested by the reviewer, we added the following statement in the revised manuscript (page 6, 1st paragraph) to discuss the change of latent heat with the change of the particle loadings:

“When the particle loading was > 0.1 wt%, we observed the decrease of the latent heat with the increase of the particle loading, which has been more commonly reported^{26, 32}. When the particle loading was <0.1 wt%, we observed the increase of the latent heat as the particle loading increased. Recently there have been several reports of the increase of the latent heat as the loading increases, especially with carbon-based nanofillers^{24, 41-47}. The favorable intermolecular interaction between paraffin and carbon nanofillers was proposed as the possible reason to account for the increase of the latent heat, but the exact mechanism is still under debate^{24, 43, 44}.”

In the revised manuscript, we also removed original ref. 43 to address the reviewer’s concern. We also want to thank the reviewer’s comment related to the Reviewer #3’s questions on the heat loss. Based on the helpful suggestions from the reviewers, we have tried to analyze in more depth of the heat loss through both experiment and theoretical calculations over the past few weeks, and provide our analysis in the following response to the Reviewer #3’s comments.

Reviewer #3 (Remarks to the Author):

- 1) We think the authors have done a good job addressing most of the comments in the previous round of review. The added discussion on the potential application in high-temperature solar thermal conversion process justifies the implication of the presented low-temperature demonstration. The added details clarify the experimental procedures and greatly help readers to interpret the results. Overall, we think the manuscript has improved a lot. Meanwhile, we have one additional suggestion regarding the heat transfer analysis for the authors to consider. After making the heat transfer analysis more rigorous, we think the manuscript is ready to be published in Nature Communications.

Reply: Many thanks for the comments and also the suggestion regarding the heat

transfer analysis. We have tried different ways to assess the heat loss through both experimental measurements and theoretical analyses over the past few weeks, and provide our analysis (after the following suggestions from the reviewer) to address the reviewer's concern.

2) Suggestion on heat transfer analysis:

The suggestion is related to comment 3 and 4 in the previous review. In comment 3, we pointed out that the reduced front surface heat loss of MOC compared to TC and OC is not enough to explain the observed improved charging efficiency. In the response, the discrepancy is explained by the difference in temperature profiles, which also lead to different temperature dependent heat losses from both the front and top surfaces. In general, we agree with the authors that the improved charging efficiency must relate to the difference in heat losses, which are ultimately determined by the temperature profiles. However, we think the exact reason that MOC performs better than TC and OC is not clearly stated from a heat transfer perspective. The added sentence in the manuscript states that the temperature profile and heat losses are different among the three charging modes. However, the reason that MOC has the minimum heat loss is not clear. From Figure 4b, MOC has the most uniform and broad temperature profile. Graphically, since heat losses are represented by the area between the measured temperature profile and the ambient temperature, it is not immediately obvious that MOC has significantly reduced heat loss compared to the other two modes. We believe the reason that MOC has the minimum heat loss is of the central importance of this work. So we suggest the authors elaborate more here.

Another related suggestion is about neglecting heat losses in the theoretical model. We understand that the actual heat loss is hard to track by the simple analytical approach. However, simple estimation can usually be used to gain useful understandings about the model. For example, as in the energy conservation in Eq. S15, an additional heat loss term can be added to the left-hand side of the equation:

$$q_{\text{loss}} = h_{\text{loss}} P \Delta x (T_i - T_{\text{amb}}) / A_{\text{cuvette}}$$

where h_{loss} is the heat loss coefficient ($\text{W}/\text{m}^2 \text{K}$), P is the perimeter of the cuvette (m), A_{cuvette} is the cuvette cross section area (m^2), T_{amb} is the ambient temperature. While the actual h_{loss} is challenging to quantify, some guesses can be made based on the insulation material property/geometry, convection correlation, and radiation. With the help of the improved model, the simulation should be closer to the experimental results (Figure 3c). In addition, the model should also shed light on the reason why MOC has better performance compared to the other modes.

Reply: Thanks again for the helpful suggestions from the reviewer. In the revised manuscript, we added the following analysis of the heat loss, the estimation of the heat loss, and the discussions of possible reasons for the minimum heat loss of MOC mode in the Supplementary Information (page S7–S12), and also added the corresponding statements in the main text (Page 6, last paragraph; page 9, 2nd paragraph; page 12, 3rd paragraph) for the references to these analysis and discussions.

1. Analysis of the heat loss

In the experimental system, as the light transmitted through the front surface of the quartz container, there will be optical losses through reflection at the surface of the quartz (~8.5%, Figure 1). For the TC mode, the reflection at the surface of the black aluminum absorber (~4%, Figure 1) further reduced the light intensity reaching the PCM system.

Figure 1. Reflection at the surfaces of the quartz container and the black aluminum absorber.

After the light was absorbed either by the black aluminum absorber in TC mode or by the PCM in the OC and MOC mode, the light energy will be converted into thermal energy (Q_{input}). Part of the Q_{input} is used to charge the PCM, which is $Q_{storage}$. The difference between Q_{input} and $Q_{storage}$ is the heat loss (Q_{loss}). As shown in Figure 2, the Q_{loss} consists of the following components:

Figure 2. Possible heat losses from the experimental system.

- (1) The heat loss from the top surface of the PCM--- $Q_{\text{loss}}^{\text{PCM_top}}$
- (2) The heat loss from the PCM to the quartz container, and such heat loss has the following components:

- (a) The heat loss that is used to heat up the container, which is $Q_{\text{loss}}^{\text{container_sensible}}$.

The heat conducted from PCM increases the temperature of the quartz container. Quartz has a relatively larger thermal conductivity ($\sim 1.3 \text{ W/m K}$) than the PCM ($\sim 0.2 \text{ W/m K}$). Heat thus can spread into much larger area in the walls of the quartz container than in the PCM. Such spreading increases the amount of quartz that is heated up during the charging process.

- (b) The heat loss from the front of the container, which is $Q_{\text{loss}}^{\text{container_front}}$.

Considering the thickness ($\sim 1.5 \text{ mm}$) and the height ($\sim 5.5 \text{ mm}$) of the quartz wall, the surface area for this heat loss ($Q_{\text{loss}}^{\text{container_front}}$) is actually much larger than the surface area of the front end of the PCM ($4 \text{ mm} \times 4 \text{ mm}$). As shown in Figure 2, the height of the quartz wall ($\sim 5.5 \text{ mm}$) is designed to be larger than the height of PCM ($\sim 4 \text{ mm}$) to prevent the spill of the PCM during the melting process, especially at high temperatures. The $Q_{\text{loss}}^{\text{container_front}}$

includes the convection and radiation heat loss from the outer front surface and also the convection and radiation heat loss from the inner front surface, both of which are exposed to air during the charging process.

- (c) Heat losses from other parts of the quartz container, which is $Q_{\text{loss}}^{\text{container_other}}$,

including the heat loss from the two sides, bottom, and backside of the

container. Considering the relatively low temperature shown at the backside of the container, we only considered the heat losses from the two sides and the bottom of the container for this portion of heat loss. The heat loss at the two sides includes the heat loss from both the outer and inner surfaces of the quartz container. The inner surfaces are exposed to air so there are convection and radiation heat losses. The two side surfaces at the outer and the bottom surface of the container are in direct contact with the thermal insulation, so the convection and radiation heat loss is suppressed, but there is still heat flow from the heated quartz surfaces to the insulation foam. We did observe the heating up of the insulation foam during the experiment (please refer to the details in the following section of “estimation of the heat losses”). With the large surface area of the side and bottom surfaces, and also the spreading of the heat in the quartz wall due to the relatively large thermal conductivity of the quartz, which further increases the surface area for the heat loss, the heat loss from the two sides and the bottom of the container is relatively substantial, so this part of heat loss is also considered in our analysis.

2. Estimation of the heat loss

Based on the above analysis of the heat loss, in this section we tried to use the following equations to estimate each part of the heat losses:

$$(1) Q_{\text{loss}}^{\text{PCM_top}}$$

Heat loss from the top surface of the charged PCM could be calculated by:

$$Q_{\text{loss}}^{\text{PCM_top}} = \int_0^t h_{\text{loss}}^{\text{PCM_top}} A_{\text{top}} (T_{\text{ave_top}} - T_{\infty}) dt' = h_{\text{loss}}^{\text{PCM_top}} \int_0^t A_{\text{top}} (T_{\text{ave_top}} - T_{\infty}) dt' \quad (1)$$

where A_{top} is the top surface area of the PCM, $T_{\text{ave_top}}$ is the average top surface temperature of the charged PCM, and T_{∞} is the ambient temperature. Here we used an average temperature ($T_{\text{ave_top}}$) and an average heat transfer coefficient ($h_{\text{loss}}^{\text{PCM_top}}$) to calculate this portion of the heat loss.

$$(2) Q_{\text{loss}}^{\text{container_sensible}}$$

The sensible heat gained by the quartz container during the heating up of the container ($Q_{\text{loss}}^{\text{container_sensible}}$) could be calculated by:

$$Q_{\text{loss}}^{\text{container_sensible}} = c_q \rho_q d A_f (T_{\text{ave_front}} - T_0) + 2c_q \rho_q d A_s (T_{\text{ave_side}} - T_0) + c_q \rho_q d A_{\text{bottom}} (T_{\text{ave_bottom}} - T_0) \quad (2)$$

where d is the thickness of the quartz wall (1.5 mm), ρ_q is the density of quartz (2.203 kg/m³), and c_q is the specific heat of quartz (0.7 J/g k). A_f (= A_{f_outer} in Figure 2), A_s (= A_{s_outer} in Figure 2) and A_b are the area of front, side and bottom surfaces of the container, respectively. T_0 is the initial temperature of the container. $T_{\text{ave_front}}$, $T_{\text{ave_side}}$ and $T_{\text{ave_bottom}}$ are the average temperature of front, side and bottom surfaces of the container, respectively. In the second term, the factor of 2 for A_s is to account for the two side walls of the container. Here we use the average temperature for each surface to estimate the sensible heat in the quartz walls.

$$(3) Q_{\text{loss}}^{\text{container_front}}$$

Heat loss from the quartz container front surface could be estimated by:

$$Q_{\text{loss}}^{\text{container_front}} = \int_0^t h_{\text{loss}}^{\text{quartz_front}} [(A_{f_outer} + A_{f_inner})(T_{\text{ave_front}} - T_{\infty})] dt' \\ = h_{\text{loss}}^{\text{quartz_front}} \int_0^t (A_{f_outer} + A_{f_inner})(T_{\text{ave_front}} - T_{\infty}) dt' \quad (3)$$

Again, we also use the average temperatures and an average $h_{\text{loss}}^{\text{quartz_front}}$ for the estimation.

$$(4) Q_{\text{loss}}^{\text{container_other}}$$

Besides the front surface, heat could also be dissipated through other parts of the container that were heated up. The heat losses from these parts are a little complex, with the inner surfaces exposed to air, while the outer surfaces are insulated. To simplify the estimation, we also used an average heat transfer coefficient ($h_{\text{loss}}^{\text{quartz_other}}$) to estimate this portion of heat loss from the side walls and the bottom of the quartz container.

$$Q_{\text{loss}}^{\text{container_other}} = \int_0^t h_{\text{loss}}^{\text{quartz_other}} [2(A_{s_outer} + A_{s_inner})(T_{\text{ave_side}} - T_{\infty}) + A_b(T_{\text{ave_bottom}} - T_{\infty})] dt' \\ = h_{\text{loss}}^{\text{quartz_other}} \int_0^t [2(A_{s_outer} + A_{s_inner})(T_{\text{ave_side}} - T_{\infty}) + A_b(T_{\text{ave_bottom}} - T_{\infty})] dt' \quad (4)$$

Based on the reviewer's suggestion to add into Equation S15 an additional heat loss term that includes an average heat loss coefficient, we used an overall average heat loss coefficient ($h_{\text{loss_ave}}$) to replace all the heat transfer coefficients in the above equations. Following are the new equations with this average heat transfer coefficient ($h_{\text{loss_ave}}$):

$$Q_{\text{loss}}^{\text{PCM_top}} = h_{\text{loss_ave}} \int_0^t A_{\text{top}} (T_{\text{ave_top}} - T_{\infty}) dt' \quad (5)$$

$$Q_{\text{loss}}^{\text{container_front}} = h_{\text{loss_ave}} \int_0^t (A_{\text{f_outer}} + A_{\text{f_inner}}) (T_{\text{ave_front}} - T_{\infty}) dt' \quad (6)$$

$$Q_{\text{loss}}^{\text{container_other}} = h_{\text{loss_ave}} \int_0^t [2(A_{\text{s_outer}} + A_{\text{s_inner}})(-T_{\infty}) + A_{\text{b}}(T_{\text{ave_bottom}} - T_{\infty})] dt' \quad (7)$$

The overall energy balance equation can thus be written as:

$$Q_{\text{input}} - Q_{\text{storage}} = Q_{\text{loss}} \quad (8)$$

$$\begin{aligned} Q_{\text{loss}} &= Q_{\text{loss}}^{\text{PCM_top}} + Q_{\text{loss}}^{\text{container_sensible}} + Q_{\text{loss}}^{\text{container_front}} + Q_{\text{loss}}^{\text{container_other}} \\ &= Q_{\text{loss}}^{\text{container_sensible}} + h_{\text{loss_ave}} \int_0^t [(A_{\text{f_outer}} + A_{\text{f_inner}})(T_{\text{ave_front}} - T_{\infty}) + 2(A_{\text{s_outer}} + \\ &A_{\text{s_inner}})(T_{\text{ave_side}} - T_{\infty}) + A_{\text{b}}(T_{\text{ave_bottom}} - T_{\infty}) + A_{\text{top}}(T_{\text{ave_top}} - T_{\infty})] dt' \end{aligned} \quad (9)$$

With the known light intensity, and the optical losses due to the reflection at the quartz surface and the surface of the black aluminum absorber, Q_{input} can thus be calculated. Q_{storage} can be estimated through the similar analysis as shown in Figure S7 in the Supplementary Information, and thus Q_{loss} can be estimated through Equation 8. From Equation 9, if we can estimate $Q_{\text{loss}}^{\text{container_sensible}}$ and obtain the average temperature of all the surfaces, we can then estimate the $h_{\text{loss_ave}}$.

To calculate $Q_{\text{loss}}^{\text{container_sensible}}$, we will need to have the average temperature for all the surfaces of the quartz container, which are challenging to obtain for the sample with the thermal insulation. We first tried to estimate $Q_{\text{loss}}^{\text{container_sensible}}$ without

thermal insulation since we could use IR camera to map out the temperature profile of all the surfaces of the quartz container, and then obtain the average temperatures for all the surfaces of the quartz container. Figure 3 shows the average temperature of the front and the side surfaces of the quartz container without the thermal insulation. With the heat spreading to the area of A_{f_outer} (the outside surface area of the quartz container), which is larger than the heat receiving surface area (front end of the PCM), the average temperature of A_{f_outer} is thus lower than the maximum top surface temperature of the PCM.

Figure 3. Average temperatures of the front surface (a) and side surface (b) of the quartz container for all three charging modes.

Figure 3a shows that the average temperature of the front surface of the quartz container for the TC mode is much higher than those for the OC mode and MOC mode. Such temperature difference not only leads to the smallest convection and radiation heat loss at the front ($Q_{loss}^{container_front}$) for the MOC mode, but also the smallest heat loss to heat up the quartz container ($Q_{loss}^{container_sensible}$) for the MOC mode. We observed the similar trend in the side surfaces of the quartz container (Figure 3b), which also leads to the smallest heat loss for the MOC mode.

With the obtained temperature profiles through IR imaging, the average temperature for all quartz surfaces can thus be calculated. We used the temperature profile in the side surfaces that was close to the bottom of the container to estimate the average temperature of the bottom of the container. The $Q_{loss}^{container_sensible}$ can thus be calculated using Equation 2 and the average heat loss coefficient (h_{loss_ave}) can then be calculated using Equation 9. The h_{loss_ave} was estimated to be $\sim 10.0 \text{ W/m}^2 \text{ K}$ for TC mode, $\sim 9.2 \text{ W/m}^2 \text{ K}$ for OC mode, and $\sim 9.1 \text{ W/m}^2 \text{ K}$ for MOC mode, respectively. With the estimated h_{loss_ave} , we also calculated the heat loss contributions from all

the quartz surfaces and the PCM using Equation 5-7, and plotted them together with $Q_{\text{loss}}^{\text{container_sensible}}$ in Figure 4. In Figure 4, $Q_{\text{loss}}^{\text{container_other}}$ in TC mode was used as the benchmark. As shown in Figure 4, for MOC mode, the heat loss that heats up the quartz container is the smallest among all three charging modes. The same is true for the heat loss from both the front surfaces and other surfaces of the quartz container. Figure 4 represents the case without the thermal insulation, but it does show that besides the heat loss from the front surface of the quartz container, there are other key heat loss mechanisms ($Q_{\text{loss}}^{\text{container_sensible}}$ and $Q_{\text{loss}}^{\text{container_other}}$) that contributes to the smaller heat loss for the MOC mode, as suggested by the reviewer. With the thermal insulation, the quartz container will be heated up more so $Q_{\text{loss}}^{\text{container_sensible}}$ will be larger than in the case without the thermal insulation. With the thermal insulation, the quartz surfaces that are heated will also heat up the polystyrene foam that is used as the thermal insulation. In the experiment, we embedded a thermocouple into the insulation foam near the front charging face (close to the same height as the center of the PCM and ~ 1.5 mm away from the quartz container outer wall) to measure the temperature of the foam during the charging process. After charging for 270 s, the insulation foam was heated up to 60 °C, 45°C and 40 °C for TC, OC and MOC mode, respectively. Considering the relatively large surface areas of the quartz container that were insulated, and also the heat spreading within the quartz wall that further increased the heated surface area of the quartz, the heat loss from the quartz surface to the insulation foam would be quite substantial as well for the system with the thermal insulation.

Figure 4. Estimation of heat losses in three different charging modes without the thermal insulation.

3. Simulation of the temperature distribution with heat losses

With the above heat loss analysis and estimation, we tried to calculate the temperature distribution by taking into consideration of the heat losses. As suggested by the reviewer, we can add the heat loss term to the Equation S15 to improve the model for the theoretically calculation. Based on the heat loss analysis discussed above, we added two heat loss terms in the Equation S15 shown below: one is related to the heat loss that heats up the quartz container, and the other is the heat loss that accounts for all other components of the heat losses.

$$\{I_0[\exp[-\alpha(i-1)\Delta x] - \exp[-\alpha i\Delta x]] - h_{\text{loss_ave}}p\Delta x(T_i - T_\infty)/A_c\}(\sum_{j=1}^n t_j - \sum_{j=1}^i t_j) - \rho_q c_q d l \Delta x (T_i - T_m)/A_c = \rho_l c_l \Delta x (T_i - T_m) \quad (10)$$

where p is the perimeter of the cuvette, A_c is the PCM section area ($0.004 \times 0.004 \text{m}^2$), T_∞ is the ambient temperature, T_m is the melting temperature of PCM, d is the thickness of the quartz container (1.5 mm), ρ_q is the density of quartz (2.203 kg/m^3), c_q is the specific heat (0.7 J/g k), l ($3 \times 0.004 \text{m}$) is the length of the quartz wall that surrounds the PCM slice. In this equation, we assume that the quartz walls are in thermal equilibrium with the PCM and the temperature of the quartz walls is same as the temperature of the PCM.

In Equation 10, we first used the estimated average heat loss coefficient ($h_{\text{loss_ave}}$) for the OC and MOC mode to calculate the temperature distribution of the PCM for the OC and MOC mode. Such average $h_{\text{loss_ave}}$ was estimated without the thermal insulation, so heat loss occurred at all the surfaces of the quartz container and also the top of PCM. To estimate the $h_{\text{loss_ave}}$ with the thermal insulation, we consider an extreme case in which the thermal insulation is an ideal thermal insulation. For such ideal thermal insulation, it does not absorb any heat from the heated quartz walls, and it prevents all the heat losses from the heated quartz walls that are in contact with such thermal insulation. In this case, besides $Q_{\text{loss}}^{\text{container_sensible}}$, the rest of Q_{loss} can only be dissipated through the exposed surfaces of the quartz container. Using such insulation, the estimated $h_{\text{loss_ave}}$ represents a high boundary of $h_{\text{loss_ave}}$. By using Equation 9, and using only the surface areas exposed to air, we estimated such high boundary of $h_{\text{loss_ave}}$ to be $\sim 20.5 \text{ W/m}^2 \text{ K}$ for TC mode, $\sim 18.2 \text{ W/m}^2 \text{ K}$ for OC mode, and $\sim 17.8 \text{ W/m}^2 \text{ K}$ for MOC mode. We used these $h_{\text{loss_ave}}$ for the OC and MOC mode to calculate the temperature distribution profiles of the PCM in the OC and MOC mode using Equation 10 as well.

In the simulation of the temperature distribution for the TC mode using Fluent, we also considered the heat loss that heats up the quartz container and the heat loss dissipates at the other parts of the system. In Fluent, these heat losses were considered in the boundary conditions. The heat loss that heats up the quartz container was calculated using Equation 2, and such heat loss ($Q_{\text{loss}}^{\text{container_sensible}}$) was accounted in the input light energy (Q_{input}) in the simulation of the temperature distribution for the TC mode. The other parts of the heat loss were considered to be dissipated at the surfaces exposed to air during the simulation using Fluent. For those other parts of the heat loss, an average heat loss coefficient $h_{\text{loss_av}}$ of 10.0 W/m² K at the lower boundary and 20.5 W/m² K at the higher boundary was used in the calculation of the temperature distribution of the PCM in the TC mode.

Figure 5 shows the calculated temperature distributions using both the low and high boundary $h_{\text{loss_ave}}$ for all three charging modes. The inset plots represent the experimentally measured temperature distributions. By considering the heat losses, the calculated temperature distributions are getting closer to the experimental measurement. In general, the temperature at the starting position calculated using the higher boundary of the estimated $h_{\text{loss_ave}}$ is smaller than the temperature measured in the experiment, while such temperature calculated using the lower boundary of the estimated $h_{\text{loss_ave}}$ is larger than the temperature measured in the experiment, which implies that the $h_{\text{loss_ave}}$ for the PCM system might be between those two estimated values.

Figure 5. The calculated and measured temperature distribution profiles for all the charging modes after 270 s.

We added the above discussions in the revised Supplementary Information (page S7–S12). It should be mentioned that the precise measurement and calculation of the exact heat losses is still challenging. We do appreciate very much the great suggestions from the reviewers, which helped us in gaining further understanding of the heat losses within the system.

Reviewers' Comments:

Reviewer #2:

Remarks to the Author:

I thank the authors for the last modifications to the article. I think they have addressed all of my concerns and I am comfortable recommending it for publication in Nature Communications.

Reviewer #3:

Remarks to the Author:

In the revised manuscript as well as the response to our previous comments, the authors have done a satisfactory job at making the heat loss analysis much more rigorous, which was the major concern in our last review. We do not have additional concern or suggestion and we believe it is ready for publication in Nature Communications.

REVIEWERS' COMMENTS:

Reviewer #2 (Remarks to the Author):

I thank the authors for the last modifications to the article. I think they have addressed all of my concerns and I am comfortable recommending it for publication in Nature Communications.

Reply: We sincerely thank the reviewer for taking time to review our work and helping improve the manuscript.

Reviewer #3 (Remarks to the Author):

In the revised manuscript as well as the response to our previous comments, the authors have done a satisfactory job at making the heat loss analysis much more rigorous, which was the major concern in our last review. We do not have additional concern or suggestion and we believe it is ready for publication in Nature Communications.

Reply: We really appreciate the reviewer for the effort in reviewing our manuscript and the helpful comments in improving the manuscript.